# Coupling framework (1.0) for the ice sheet model PISM (1.1.4) and the ocean model MOM5 (5.1.0) via the ice-shelf cavity module PICO in an Antarctic domain

Moritz Kreuzer[1,2], Ronja Reese[1], Willem Nicholas Huiskamp[1], Stefan Petri[1], Torsten Albrecht[1], Georg Feulner[1], and Ricarda Winkelmann[1,2]

[1]Earth System Analysis, Potsdam-Institute for Climate Impact Research (PIK), Member of the Leibniz Association, P.O. Box 60 12 03, 14412 Potsdam, Germany
[2]University of Potsdam, Institute of Physics and Astronomy, Karl-Liebknecht-Str. 24-25, 14476 Potsdam, Germany

**Correspondence:** Moritz Kreuzer (kreuzer@pik-potsdam.de), Ricarda Winkelmann (ricarda.winkelmann@pik-potsdam.de)

**Abstract.** The past and future evolution of the Antarctic Ice Sheet is largely controlled by interactions between the ocean and floating ice shelves. To investigate these interactions, coupled ocean and ice sheet model configurations are required. Previous modelling studies have mostly relied on high resolution configurations, limiting these studies to individual glaciers or regions over short time scales of decades to a few centuries. We present a framework to couple the dynamic ice sheet model PISM with the global ocean general circulation model MOM5 via the ice-shelf cavity module PICO. Since ice-shelf cavities are not resolved by MOM5, but parameterised with the box model PICO, the framework allows the ice sheet and ocean components to be run at resolutions of $16\,\mathrm{km}$ and $3°$, respectively. This approach makes the coupled configuration a useful tool for the analysis of interactions between the Antarctic Ice Sheet and the global ocean over time spans on the order of centuries to millennia. In this study we describe the technical implementation of this coupling framework: sub-shelf melting in the ice sheet component is calculated by PICO from modeled ocean temperatures and salinities at depth of the continental shelf and, vice versa, the resulting mass and energy fluxes from melting at the ice-ocean interface are transferred to the ocean component. Mass and energy fluxes are shown to be conserved to machine precision across the considered component domains. The implementation is computationally efficient as it introduces only minimal overhead. Furthermore, the coupled model is evaluated in a 4000 year simulation under constant present-day climate forcing and found to be stable with respect to the ocean and ice sheet spin-up states. The framework deals with heterogeneous spatial grid geometries, varying grid resolutions and time scales between the ice and ocean component in a generic way, and can thus be adopted to a wide range of model setups.

## 1 Introduction

Most of Antarctica's coastline is comprised of floating ice shelves where glaciers of the Antarctic Ice Sheet drain into the surrounding Southern Ocean. Mass loss of these ice shelves occurs through ocean-induced melting at their base and calving

of icebergs which both contribute about the same amount (Depoorter et al., 2013). Observations show that ice shelf-ocean interaction has been the main driver for mass loss of the West Antarctic Ice Sheet for the past 25 years (Jenkins et al., 2018; Shepherd et al., 2018; Holland et al., 2019). Ocean forcing has also been identified as playing a major role in past changes of the Antarctic Ice Sheet. Evidence that the Holocene retreat of the West Antarctic Ice Sheet was driven by warm water intrusions onto the continental shelf was provided by paleo proxy data analysis of Hillenbrand et al. (2017) and supported by ensemble modelling for the Ross Embayment (Lowry et al., 2019). Ice sheets respond to changing oceanic and atmospheric conditions, but they also feed back to the Earth's climate in various ways, e.g. through meltwater input into the oceans, sea level change or change of atmospheric circulation and precipitation patterns resulting from changes in orography and albedo (Nowicki et al., 2020; Vizcaíno et al., 2014). To study interactions and feedbacks between the Antarctic ice sheet and the ocean, e.g., through melt-induced freshwater input into the ocean, numerical models are an important tool. As the large ice sheets have long response timescales, coupled simulations over millennia are necessary to capture long-term effects. Such coupled simulations are also useful to study the long-term past or future evolution of ice-sheets and oceans. This, together with the advantage of using ensemble simulations to constrain uncertainty in parameterised processes, makes computational efficiency a key requirement for such coupled models.

Existing land ice-ocean modelling approaches can be classified in five major categories which will be briefly introduced below:

1. global ocean/atmosphere models with fixed ice sheets

2. standalone ice sheet models with simplified ocean forcing

3. high resolution ocean models resolving ice shelf cavity geometries

4. high resolution, regional coupled ice-ocean models

5. global, coarse grid ice-ocean coupled models with simplified ice-ocean interactions.

The standard set of experiments for the Coupled Model Intercomparison Projects (CMIP) are performed by Atmosphere-Ocean General Circulation Models which use fixed, non-dynamic ice sheet configurations and have thus only a limited representation of the aforementioned interactions and feedbacks (category 1; e.g. Eyring et al., 2016). CMIP-style models are computationally demanding which usually limits their application to centennial time scales (Balaji et al., 2017). For transient runs beyond the 21st century, however fixed ice sheets would be an unrealistic assumption.

Ice dynamics missing in standalone climate models are traditionally computed by likewise standalone ice sheet models (category 2) as ice dynamics typically respond on centennial to millenial time scales. Those simulations rely on external forcing, most notably for atmospheric and oceanic boundary conditions. Ocean forcing is applied either through prescribed melt rates or through parameterisations of various complexity based on temperature, salinity or pressure, for example; see Asay-Davis et al. (2017) for a more in-depth discussion. The latter approach is for instance used in the Ice Sheet Model Intercomparison Project for CMIP6 (ISMIP6; Nowicki et al., 2020; Seroussi et al., 2020; Jourdain et al., 2020), where standalone ice-sheet

models are forced by atmospheric and oceanic boundary conditions from CMIP5 (Taylor et al., 2012) to constrain Antarctic mass loss and sea-level rise until the end of the century. The low computational cost of melt parameterisations for standalone ice sheet models allows experiments to be integrated on multi-millennial time-scales. However, this comes with uncertainties in oceanic boundary conditions not only due to the absence of a dynamic ocean, but also due to missing feedbacks between ice and ocean.

A much more detailed representation of the ice-ocean boundary layer processes is achieved with high resolution, cavity resolving ocean models (category 3). Usually, this model type simulates the ice shelf geometry as static but thermodynamically active (e.g. Donat-Magnin et al., 2017). Their application ranges from idealised-geometry setups to specific regions like the Weddell or Amundsen Sea and even circum-Antarctic setups. High-resolution ocean modelling (horizontal resolution in the order of 1-10 km) is needed to capture the complex processes determining the water masses that access the ice shelf cavities and the amount of heat that is available for melting the ice. A detailed discussion of these processes including a list of available models is given in Dinniman et al. (2016).

Closely related to ice-shelf cavity resolving ocean models are coupled ice-ocean high resolution models (category 4), which include an additional representation of grounded and floating ice dynamics. These models have been applied to idealised geometries (e.g. De Rydt and Gudmundsson, 2016) or regional set-ups (e.g. Naughten et al., 2021; Seroussi et al., 2017; Timmermann and Goeller, 2017). They can also can be used to assess simple melt parameterisations from category 2, e.g. Favier et al. (2019). While the detailed representation of sub-shelf processes is important for realistic estimates of melt rates, these highly-resolved configurations are, because of their computational demand, not practical to examine long-term and global effects of ice-ocean interaction.

This is however crucial because including freshwater fluxes from the Antarctic Ice Sheet in simulations of Global Circulation Models has been shown to influence global ocean temperatures and their variability, precipitation patterns as well as to increase Antarctic ice loss through trapping warm water below the sea surface (Bronselaer et al., 2018; Golledge et al., 2019). To study these effects on long timescales, a relatively new type of models is useful: large-scale ice-ocean models coupled via simplified melt parameterisations (category 5). Examples for global ocean-ice sheet coupling approaches are Goelzer et al. (2016) and Ziemen et al. (2019). Both use melt parameteristaions which describe the melt process directly at the ice-ocean interface (Beckmann and Goosse, 2003; Holland and Jenkins, 1999). In addition to the melting at the ice-ocean interface, the Potsdam Ice-shelf Cavity mOdule (PICO; Reese et al., 2018) mimics the large-scale overturning circulation in ice shelf cavities. PICO can model melt rates in accordance with observations (Rignot et al., 2013): while in cold cavities, e.g., underneath Filchner-Ronne Ice Shelf, average melt rates are at the order of 1 m/a, melt rates in warm cavities, as found in the Amundsen Sea, are at the order of 10 m/a. At the same time PICO is computationally efficient compared to high resolution, cavity-resolving ocean models. So far PICO has been used for standalone ice sheet modelling (category 2 from above) e.g. in Reese et al. (2020) and Albrecht et al. (2020), but as PICO is driven by far-field ocean temperature and salinity in front of the ice shelf cavities, it can also act as a coupler between non-cavity resolving ocean models and ice sheet models.

To study the ice sheet and ocean system on global and multi-millennial scale, we present a category 5 framework for the dynamical coupling of the Parallel Ice Sheet Model (PISM; Bueler and Brown, 2009; Winkelmann et al., 2011) and a coarse

resolution configuration of the Modular Ocean Model (MOM5; Griffies, 2012) using PICO. The design of the presented framework follows three criteria: (1) mass and energy conservation needs to be ensured over both ocean and ice sheet component domains, (2) the coupling framework should not introduce a performance bottleneck to the existing standalone models and (3) the framework should follow a generic and flexible design independent of specific grid resolutions or number of deployed CPUs.

In the following we introduce the ice-sheet and ocean components in use, including their grid definitions (Section 2). The framework design including the variables that are exchanged between the components is discussed in Section 3, followed by a detailed description of inter-component data processing in Section 4. The framework's computational performance, conservation of mass and energy and results of coupled simulations for present-day conditions are evaluated in Section 5, followed by a discussion (Section 6) and conclusions (Section 7).

## 2 Models

The following paragraphs introduce the ice-sheet model PISM including its sub-shelf cavity module PICO and the ocean model MOM5, which are coupled as components into the framework.

### 2.1 Ice sheet model PISM and the ice-shelf cavity module PICO

The Parallel Ice Sheet Model[1] (PISM) is an open-source model which simulates ice sheets and ice shelves using a finite-difference discretisation (Bueler and Brown, 2009; Winkelmann et al., 2011). PISM is defined on a regular Cartesian grid as shown in Fig. 1a, which is projected on a WGS84 ellipsoid (Slater and Malys, 1998) or related geometries like a perfect sphere. In this work PISM is used with a horizontal resolution of $16 \, \text{km} \times 16 \, \text{km}$ with 80 vertical levels (Albrecht et al., 2020). The vertical resolution increases from $130 \, \text{m}$ at the top of the domain to $20 \, \text{m}$ at the (ice) base, with a domain height of $6000 \, \text{m}$. PISM uses a hybrid of the Shallow-Ice Approximation (SIA) and the two-dimensional Shelfy-Stream Approximation of the stress balance (SSA, MacAyeal, 1989; Bueler and Brown, 2009) over the entire Antarctic Ice Sheet. The grounding line position is determined using hydrostatic equilibrium, with sub-grid interpolation of the friction at the grounding line (Feldmann et al., 2014). PISM is a thermomechanically coupled (polythermal) model based on the Glen–Paterson–Budd–Lliboutry–Duval flow law (Aschwanden et al., 2012). The three-dimensional enthalpy field can evolve freely for given boundary conditions. We apply a power law for sliding with a Mohr–Coulomb criterion relating the yield stress to parameterised till material properties and the effective pressure of the overlaying ice on the saturated till (Bueler and van Pelt, 2015). Basal friction and sub-shelf melting are linearly interpolated on a sub-grid scale around the grounding line (Feldmann et al., 2014). The calving front position can evolve freely using the eigen-calving parameterisation (Levermann et al., 2012) which is combined with the removal of ice that is thinner than $50 \, \text{m}$. The numerical time-stepping scheme is adaptive and based on the Courant-Friedrichs-Lewy (CFL) condition among others (Bueler et al., 2007), which results in a range of time steps from minutes to years depending on the physical state of the model. The PISM source code is written in C++.

---

[1]see https://pism-docs.org/ (last accessed: April 16, 2021)

The Potsdam Ice-shelf Cavity mOdel (PICO) calculates sub-shelf melt rates and is implemented as a submodule of PISM
(Reese et al., 2018). It parameterises the vertical overturning circulation in ice-shelf cavities driven by the ice-pump mechanism,
as described by Lewis and Perkin (1986). This circulation induces melting and freezing below the ice shelves as sketched in
Fig. 3. PICO uses a box representation below the ice shelves developed by Olbers and Hellmer (2010) and extends their
approach to two horizontal dimensions. Input for PICO are ocean temperature and salinity at the depth of the continental
shelf. The strength of the overturning circulation is calculated in PICO from the density difference between the inflowing
water masses and the water masses in the first box close to the grounding line and scaled with a continent-wide overturning
coefficient, which is an internal PICO parameter. Velocities of water masses flowing into the ice-shelf cavities are therefore not
required.

## 2.2 Ocean model MOM5

The ocean component in use for this coupling setup is the Modular Ocean Model v5[2] (MOM5; Griffies, 2012) which is an
open-source, three-dimensional Ocean General Circulation Model. It is coupled via the Flexible Modelling System (FMS)
coupler to the Sea Ice Simulator (SIS; Winton, 2000). In this work, we also include SIS and FMS when referring to MOM5.
For this study, MOM5 is used with a global coarse grid setup from Galbraith et al. (2011, see Fig. 1b): the lateral model grid
is $3°$ resolution in longitude (120 cells) and in latitude it varies from $3°$ at the poles to $0.6°$ at the equator (80 cells). It makes
use of a tripolar structure to avoid the grid singularity at the North Pole (Murray, 1996). The vertical grid is defined using the
re-scaled pressure coordinate ($p^*$) with a maximum of 28 vertical layers. The uppermost eight layers are approximately $10\,\mathrm{m}$
thick, gradually increasing for deeper cells to a maximum of ca. $511\,\mathrm{m}$. The vertical resolution in depths relevant for ice shelf
cavities is between 50 and $180\,\mathrm{m}$. The lowermost cells can have a reduced thickness to account for ocean bathymetry with
partial cells. The ocean grid is not defined in the center of the Antarctic continent (south of $\approx 78°\mathrm{S}$, see Fig. 1b). The ocean-sea
ice system time steps are set to 8 hours. MOM5, SIS and FMS are written in Fortran.

## 3 Coupling Approach

The design of the coupling between the ice-sheet component PISM and the ocean component MOM5 is shown in Fig. 2,
including the exchanged variables. PICO uses two dimensional (horizontal) input fields, namely temperature and salinity of
water masses that access the ice-shelf cavities, to calculate melting and refreezing at the ice-ocean interface, as illustrated in
Fig. 3. Fluxes describing basal melt, surface runoff and calving in the ice component are used to determine the mass as well as
energy fluxes received by the ocean component.

The time scales of physical processes as well as the numerical time steps in MOM5 (hours) and PISM (years) differ by
several orders of magnitude. This is one motivation among others to use an *offline sequential coupling* approach to exchange
the fields between the two components. In this case both components are run in alternating order for the same model time, which
will be referred to as the *coupling time step*. This technical procedure is illustrated in Fig. 4. An alternative *online coupling*

---

[2]see https://mom-ocean.github.io/ (last accessed: April 16, 2021)

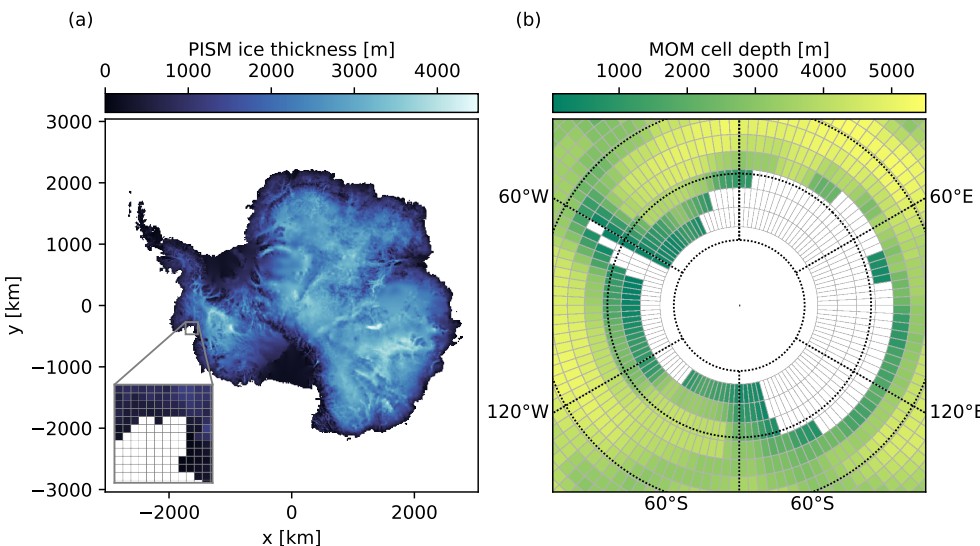

**Figure 1.** Ice-sheet and ocean component grids. (a) Ice thickness in Antarctica on the Cartesian PISM grid. The inset shows the grid structure in a coastal area for a resolution of 16 km. (b) Depth of MOM5 cells displayed in stereographic projection centered at the South Pole. Resolution at $70°$S is $\sim 3°$lat$\times 3°$lon ($\sim 330$ km$\times 115$ km). White cells are considered land by MOM5. The ocean grid extends to $78°$S. Interlocking of PISM and MOM5 domains is shown in Fig. 3 and 6a.

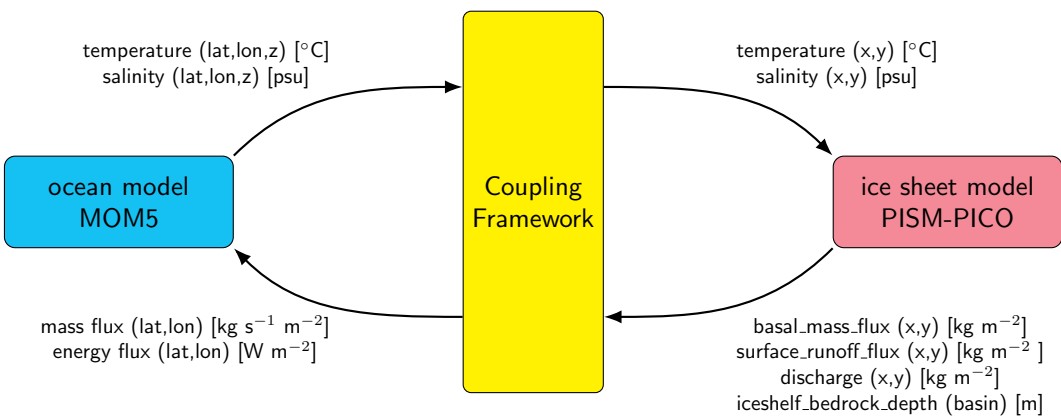

**Figure 2.** Overview of the coupling framework showing the input and output variables for the ocean component MOM5 and the ice-sheet component PISM. Dimensions of variables are given in parentheses, units in square brackets. The (lat,lon) coordinates refer to the spherical ocean component grid and the (x,y) coordinates to the Cartesian ice sheet component grid.

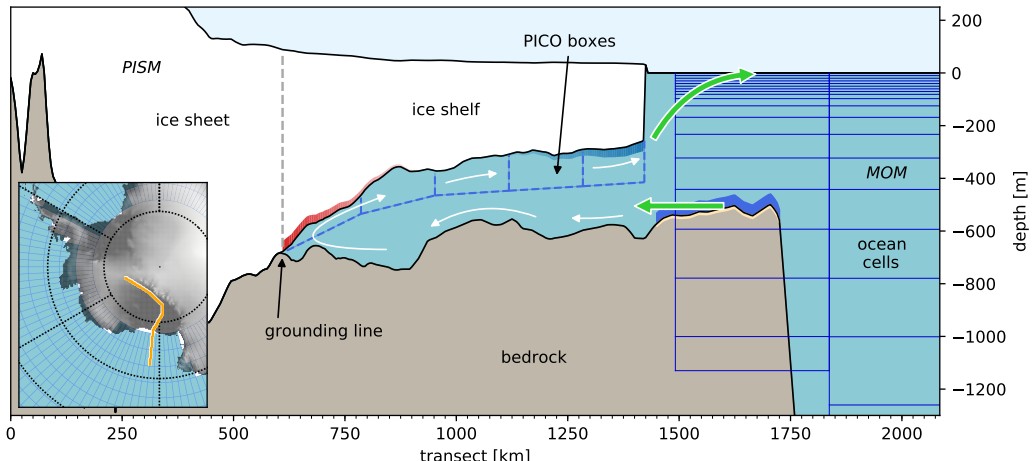

**Figure 3.** Coupling framework for the ice-sheet component PISM and the ocean component MOM5 via the ice shelf cavity module PICO. A cross section of PISM bedrock (brown) and ice thickness (white) is compared to the MOM5 ocean cells (blue continuous lines). The inset shows the transect line in orange colour in the Antarctic region. PICO boxes (blue dashed lines) follow the overturning circulation in the ice-shelf cavity. The circulation is indicated by white arrows with highest melting in the deepest regions close to the grounding line (red shade) and lower melting or refreezing in the shallower areas towards the ice shelf front (blue shade). The exchange of variables and fluxes between the two components is indicated by green arrows: PICO input from MOM5 is taken at the depth of the continental shelf (dark blue regions). Mass and energy fluxes from PICO are transferred to MOM5 through the surface runoff interface.

approach is discussed in Section 6. In the offline coupling procedure, one component is first run for the period of a coupling time step. Then the output is processed and provided as input or boundary condition to the other component, respectively. Using the modified input, the components are restarted from their previous computed state. For example, MOM5 runs for 10 years and writes annual mean diagnostics fields of temperature and salinity. PISM receives the temporal average of these fields over the coupling time step as boundary conditions for PICO, and is then integrated for the same 10 year period. Melt water

and energy fluxes derived from PISM output are aggregated over the coupling time step. The resulting fluxes are then added as external fluxes to the ocean over the course of the next integration period. To avoid shocks in the forcing, they are distributed uniformly over the entire coupling time step.

The coupling framework consists of a Bash script which implements the coupling procedure indicated in Fig. 4, making use of the software tools Climate Data Operator (CDO; Schulzweida, 2019) and netCDF Operator (NCO; Zender, 2018). The

160 output processing between the different component executions is implemented in Python scripts. Their functionality will be explained in the next Section. The code is made available in a public archive[3], see also the "Code and data availability" section below.

---

[3]https://doi.org/10.5281/zenodo.4692679 (last accessed: April 16, 2021)

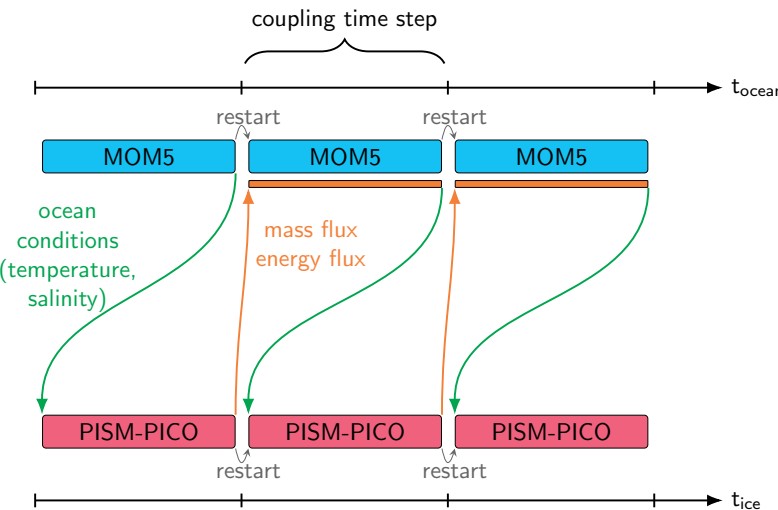

**Figure 4.** Offline coupling procedure for the PISM-MOM5 setup: the components are run sequentially for the same coupling time step and after each run, variables are exchanged. Temperature and salinity variables from the ocean component MOM5 are used as input fields for the ice component PISM-PICO. Mass and energy fluxes from PISM-PICO output are uniformly applied over the next coupling time step as input to MOM5.

## 4   Inter-Component Data Processing

To make the output of the ocean component compatible with the input requirements of the ice component and vice versa, processing of data output fields like regridding, adjustment of dimensions, unit conversion or filling of missing values is required, which is described in this section.

The ice and ocean components operate on independent, non-complementary computational grids. The inset of Fig. 3 shows that there are both, spatial gaps and overlaps, between the ocean grid cells and the ice extent represented by PISM. As the ocean grid is much coarser than the ice grid and MOM5 cells are either defined entirely as land or ocean (no mixed cells allowed), inconsistencies in the exchange of quantities between the two grids are unavoidable, requiring careful consideration of data regridding. The grid remapping mechanisms presented in the following sections are independent of the used grid resolutions.

### 4.1   Ocean to Ice

PICO uses a definition of ocean basins around the Antarctic Ice Sheet which encompass areas of similar ocean conditions at the depth of the continental shelf (Reese et al., 2018). They are based on Antarctic drainage basins defined in Zwally et al. (2012) and extended to surrounding ice shelves and the Southern Ocean, see Fig. 5b. Oceanic fields of temperature and salinity are averaged over the continental shelf for each basin and provided as input to PICO. Note that PICO uses one value of temperature and salinity per basin. Three steps are needed to process the oceanic output fields to make them usable as input to PISM:

- First, the three dimensional output fields (temperature, salinity) are remapped bilinearly from the spherical ocean grid to the Cartesian ice grid. Bilinear regridding is chosen to allow for a smooth distribution of the coarse ocean cell quantities on the finer ice grid. Only regions with valid ocean data are filled on the ice grid, which is up to the cell center of the southernmost ocean cell. Areas with missing data need to be filled accordingly (compare grey areas in Fig. 5a for example), which is done in the next step. Another option - linear extrapolation into areas with no ocean data coverage by the bilinear regridding scheme - is not applied here as it can lead to unrealistic results.

- Secondly, missing values are filled with appropriate data, namely the average over all existing values that are adjacent to grid cells with missing values. This procedure is conducted for each basin and vertical layer, using the same mean value of adjoining valid cells for all missing grid cells in that basin. Now, the continental shelf area between the ice shelf front and the continental shelf break (see Fig. 5a), which is used by PICO to calculate the basin mean values of oceanic boundary conditions, is entirely filled with appropriate values.

- Lastly, the three dimensional variables are reduced to two dimensional PICO input fields which represent the ocean conditions at the depth of the continental shelf. This is done by vertical linear interpolation: for every horizontal grid point, the data is interpolated to PISM's mean continental shelf depth of the corresponding basin. In case the ocean bathymetry is shallower than the continental shelf depth as seen by PISM, the lowermost ocean layer is chosen. An example of the processed input data for PICO is shown in Fig. 5b.

## 4.2  Ice to Ocean

To transfer the mass and energy fluxes from the ice component to the ocean component, a mapping from the PISM to the MOM5 grid is required. There are large areas of the PISM domain that are not overlapping with valid MOM5 ocean cells (see white areas in Fig. 1b and inset in Fig. 3). To ensure mass and energy conservation, we introduce a new mechanism for the coupled system which maps every southernmost ocean cell of the MOM5 grid to exactly one PICO basin (see Fig. 6). The mechanism selects the basin that the center of the MOM5 cell lies in. As one basin is usually linked to multiple ocean cells, the link proportion between each basin and their corresponding ocean cells is scaled by the ocean cell areas. An example for PISM mass fluxes and their distribution onto the MOM5 grid is shown in Fig. 7.

The mass and energy fluxes from PISM output are calculated and distributed in the following manner:

- The PISM output variables describing the surface runoff, basal mass fluxes and discharge through calving are added up. As they are given in units of $\mathrm{kg\,m^{-2}\,yr^{-1}}$, multiplication with the PISM grid cell areas and division by number of seconds per year transforms the consolidated mass flux into units of $\mathrm{kg\,s^{-1}}$.

- The energy flux from ice to ocean is obtained by multiplying the mass flux resulting from basal melt and discharge with the enthalpy of fusion ($L = 3.34 \cdot 10^5\,\mathrm{J\,kg^{-1}}$) to account for the energy required during the phase change from frozen to liquid state or vice versa. At this point the energy flux is in the unit $\mathrm{W}$. Potential diffusive heat fluxes from the ocean into

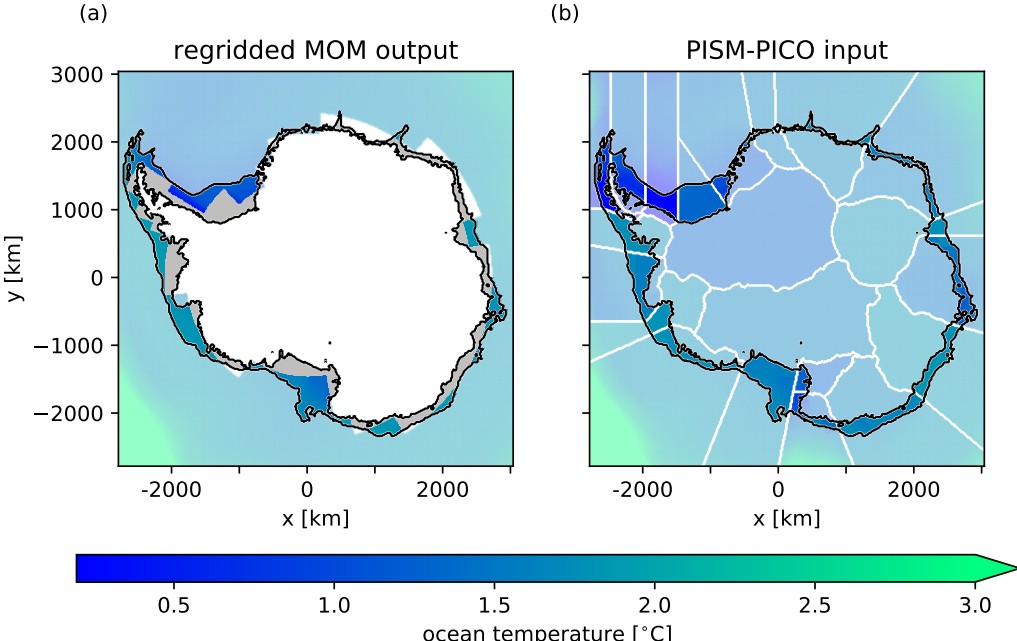

**Figure 5.** Visualisation of inter-component data processing from (a) regridded ocean component output to (b) ice component input. In (a) an example for the ocean temperature field at a depth of approximately 500 m is shown, with black contour lines indicating the continental shelf between the ice shelf front and the continental shelf break (-2000 m) as used in PICO. Missing values within that area are coloured in grey. Ocean values outside the continental shelf are not used for averaging basin mean values in PICO and therefore shown in lighter colours. The result of the processing procedure is the two dimensional ocean temperature field shown in (b), which is obtained through vertical interpolation of the filled fields applied to appropriate basin depths. PICO basins are indicated by white contour lines.

the ice as well as the energy required to warm the melt water to ambient temperatures are comparatively small (Holland and Jenkins, 1999) and thus neglected here.

– Having calculated bulk mass and energy fluxes, they can be aggregated for each PICO basin and distributed to the corresponding ocean cells with the mapping mechanism described above. On the ocean grid the fluxes are divided by the given grid cell area resulting in unit $\mathrm{kg\,s^{-1}\,m^{-2}}$ for mass and $\mathrm{W\,m^{-2}}$ for energy fluxes. These fluxes are input into the ocean surface through MOM5's internal FMS coupler.

## 5 Evaluation

In this section, the coupling setup will be evaluated on the basis of runtime performance and numerical accuracy. Physical evaluation of the coupled setup is provided for present-day conditions. Further validation and implications in terms of possible feedback mechanisms will be studied in detail in a separate article.

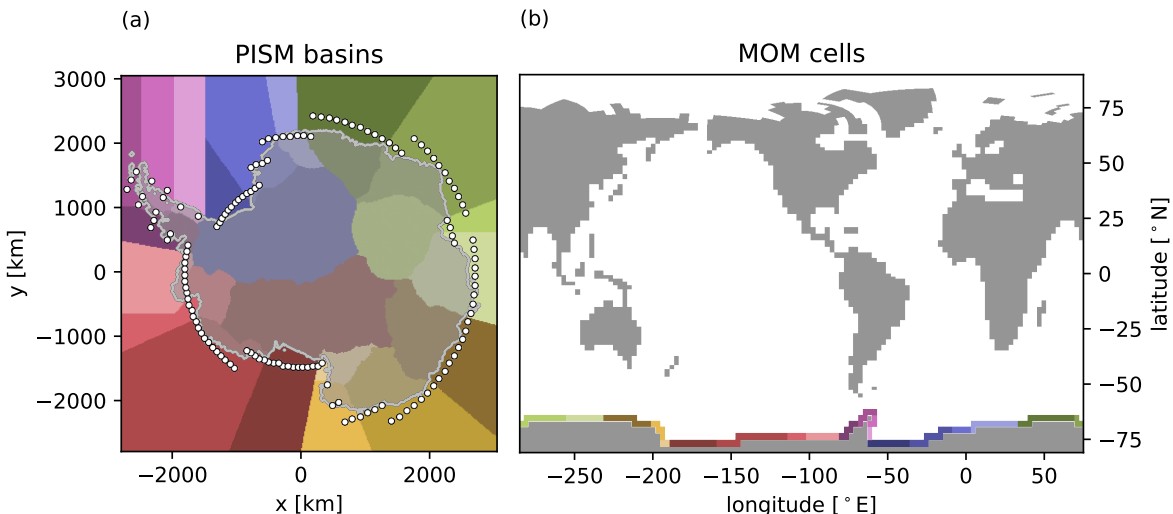

**Figure 6.** Visualisation of mapping mechanism between (a) PICO basins and (b) MOM5 ocean cells. PICO basins on the ice-sheet grid are shown in (a), with each basin assigned a different colour. The location of the centre of southernmost ocean cells is denoted by white circles. As a spatial reference, the ice cover modelled by PISM is shown in grey. Panel (b) shows the MOM5 land-ocean mask with corresponding PICO basin colours for the southernmost ocean cells surrounding the Antarctic Ice Sheet. Grey cells are considered as land in MOM5.

## 5.1 Coupled Benchmarks

The coupling framework presented here provides the tools for coupled ice sheet-ocean simulations on centennial to millennial time scales, which requires reasonably fast execution times. In the following, we analyse the coupled execution time and evaluate the efficiency of the coupling framework, using a total model runtime of 200 years on 32 cores (2 CPU nodes, each equipped with 2×8 core Intel E5-2667 v3). For the modelling of ice-ocean interactions, the coupling time step is an important parameter that requires careful adjustment, while keeping the different time scales of ice and ocean processes in mind. Too short time steps certainly yield a waste of compute time and disk space for restart and coupling overhead. Too long time steps could possibly yield instabilities and lead to a less accurate representation of ice-ocean interaction processes. Here, only the influence of the coupling frequency on the overall runtime performance is assessed, leaving the examination of physical implications to Section 5.3. Two experiments with time steps of 1 and 10 years are compared, with a total number of 200 and 20 coupling iterations, respectively. The individual coupled component simulations start from quasi-equilibrium conditions.

The elapsed total runtime (wall-clock time) required for 200 years model time is 21 976 s and 13 245 s with a coupling time step of 1 and 10 years, respectively. Figure 8 shows the runtime required for each of the individual components within the coupling framework, corresponding numbers are listed in Table A1. With a 10 year coupling time step, the core runtime of MOM5 (93%) including necessary postprocessing (2%) requires the biggest share of total runtime in the coupled setup.

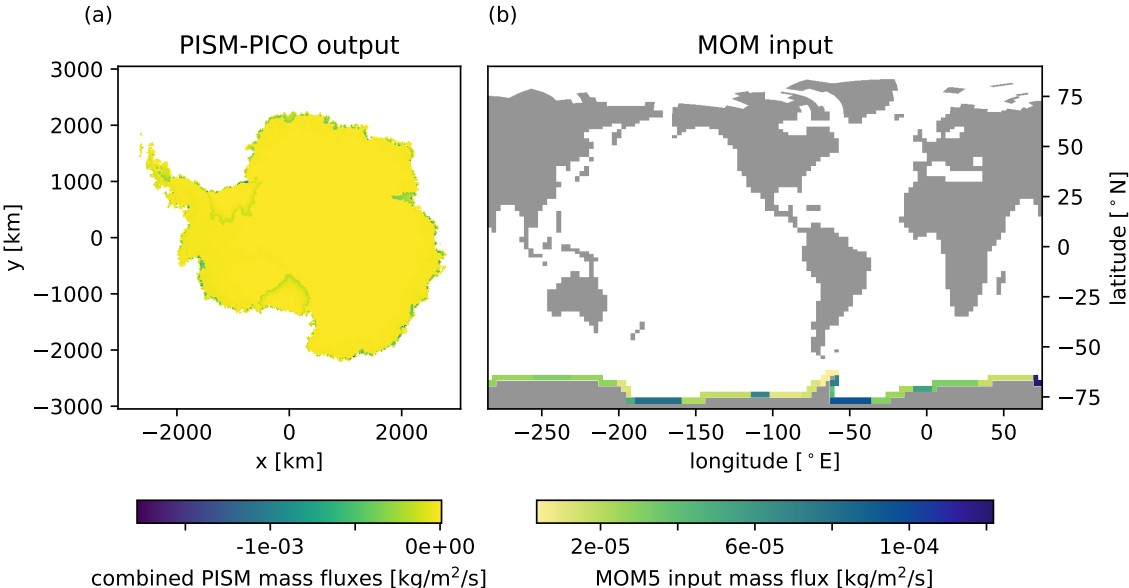

**Figure 7.** Visualisation of (a) PISM mass flux distribution to (b) the MOM5 ocean grid. PISM output variables describing surface runoff, basal melting and calving are aggregated over space and time (coupling time step) to calculate mass and energy fluxes which are processed as input to the MOM5 ocean component as described in Section 4.2. Panel (b) shows the corresponding mass flux distribution on the MOM5 grid.

The PISM runtime (4%) as well as the time needed for the coupling preprocessing (<1%) and intercomponent processing
(<2%) routines are almost negligible. This means that in the given setup, coupling the ice sheet component PISM to the ocean component MOM5, comes with minimal overhead compared to standalone ocean simulations, when using a coupling time step of 10 years.

In the experiment using a yearly coupling time step, the elapsed time for all MOM5 executions increases slightly (15 446 s) compared to 10 yearly coupling (12 267 s). The increase is due to component initialisation overhead which occurs 10 times as
often as in the decennial coupling configuration. The ocean component postprocessing (9%) and intercomponent processing routines (4%) are taking a bigger share of the total runtime as the number of executions has similarly increased by the factor 10. PISM runtimes are about 6 times greater for yearly coupling (13% of total runtime), although the total integration period in PISM is the same in both experiments. This is due to the component initialisation as well as reading and writing of input/output and restart files dominating the PISM execution of 1 model year, which is reasonable as PISM is designed, and usually used, for
much longer integration times. Overall, the total execution time increases by about 66% in the yearly coupled setup compared to the run with a coupling time step of 10 years.

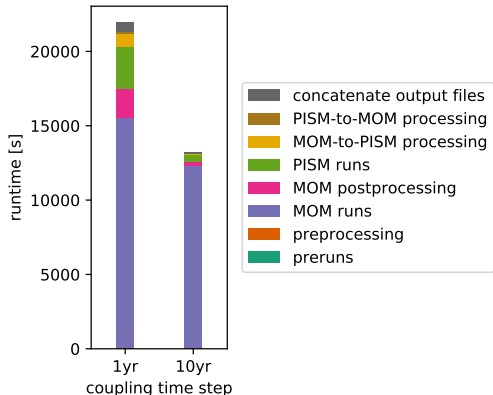

**Figure 8.** Runtimes of coupled PISM-MOM5 setup for 200 years model time, using 32 cores and coupling time steps of 1 and 10 years. PISM runtimes include PICO and MOM5 runtimes include SIS and FMS components. The elapsed time for individual components of the coupling framework is aggregated and stacked in the same order as in the legend. The runtimes are listed in Table A1.

## 5.2 Energy and mass conservation

In a coupled model, conservation of mass and energy is important to ensure that no artificial sources or sinks of these quan-
250 tities are introduced through the coupling mechanism. This is especially important in the context of paleo modelling, where simulations can span tens of thousands of years. In the presented ice-ocean coupling framework, prescribed fluxes are applied at the open system boundaries, e.g. precipitation from the atmosphere to ice and ocean or river runoff from land to ocean. To check that the total amount of mass and energy stocks is constant in the coupled system over the model integration, we assess virtual quantities. Those are obtained by subtracting the masses applied through surface fluxes from the total mass and energy
stocks calculated in the model (see Eq. (1) for mass). If the virtual model mass across the model components $m_\mathrm{v}$ is constant with fluctuations in the order of machine precision, as denoted in Eq. (2), conservation of mass is achieved.

$$m_\mathrm{v} = (m_\mathrm{o} + m_\mathrm{si} - m_\mathrm{osi}^\mathrm{s} - m_\mathrm{osi}^\mathrm{d}) + (m_\mathrm{li} - m_\mathrm{li}^\mathrm{s}) \tag{1}$$

$$\frac{d}{dt}m_\mathrm{v} \sim 0\,\mathrm{Gt/a} \tag{2}$$

The masses of the ocean, sea ice and land ice components are represented by $m_\mathrm{o}, m_\mathrm{si}$ and $m_\mathrm{li}$ respectively, while $m_\mathrm{osi}^\mathrm{s}$ and $m_\mathrm{li}^\mathrm{s}$
denote the cumulative, spatially integrated surface mass balance flux of the ocean-sea ice component MOM5/SIS and the land ice component PISM, respectively. The internal model drift of mass in the coarse-grid MOM5/SIS setup is described by $m_\mathrm{osi}^\mathrm{d}$ ($\approx 4 \cdot 10^{15}\,\mathrm{kg}$ accumulated over 200 years) and needs to be considered in the computation of virtual model mass in Eq. (1). All terms in Eq. (1) are quantities of mass with the temporal resolution of the coupling time step. The relative mass conservation error $e_\mathrm{rel}^\mathrm{m}$ is calculated as fluctuations of the virtual model mass compared to its temporal mean $\overline{m_\mathrm{v}}$, noted in Eq. (3).

$$e_{\text{rel}}^{\text{m}} = \frac{m_{\text{v}} - \overline{m_{\text{v}}}}{\overline{m_{\text{v}}}} \tag{3}$$

The relative mass conservation error $e_{\text{rel}}^{\text{m}}$ is shown in Fig. 9a for 200 model years with a yearly coupling time step. Regarding the order of magnitude of land ice mass $\mathcal{O}(m_{\text{li}}) = 10^{19}\,\text{kg}$ which is given in single precision ($\approx 7$ decimal digits) output format and the order of magnitude of ocean and sea ice mass $\mathcal{O}(m_{\text{o}} + m_{\text{si}}) = 10^{21}\,\text{kg}$, given in double precision ($\approx 16$ decimal digits) format, the shown fluctuations in the order of $10^{-9}$ are reasonable. As the relative mass error does not show a trend, no systematic error is introduced through the coupling procedure. In Fig. 9b the fluctuations of virtual model mass is also compared to the mass flux between the land ice and ocean component ($m_{\text{x}}$), which is in the order of $\mathcal{O}(10^{-3})$.

As PISM does not provide diagnostic variables to record incoming and outgoing energy fluxes across its modelled boundaries, an analysis of the total amount of enthalpy in the coupled ice-ocean system could not be easily derived. However, it is possible to show that no systematic error is induced during remapping the energy flux from PISM to MOM5 grid. Figure 9c shows the relative energy flux remapping error of the test run undertaken in Section 5.1, which is in the order of double machine precision $\mathcal{O}(1e^{-16})$.

## 5.3 Coupled runs for present-day conditions

Here we present a 4000 year (4 kyr) simulation of the coupled system under constant climate forcing for validation of the model. MOM5-SIS are forced by present-day, monthly mean fields for radiation, precipitation, surface air temperature, pressure, humidity and winds as described in Griffies et al. (2009) with an internal coupling time step of 8 hours between ocean and sea ice sub-components. River runoff from land in Antarctica is replaced by PISM fluxes. PISM is initialized from Bedmap2 geometry (Fretwell et al., 2013), with surface mass balance and surface temperatures from RACMOv2.3p2 averaged between 1986–2005 (van Wessem et al., 2018). Geothermal heat flux is from Shapiro and Ritzwoller (2004). In the spin-up of PISM, PICO is used to calculate basal melt rate patterns underneath the ice shelves and driven by observed ocean temperature and salinity values on the continental shelves (1975–2012, Schmidtko et al., 2014).

Spin-up states for ocean and ice models are computed separately prior to coupling for 10 kyr and 210 kyr, respectively. To reduce a shock by changes in the river runoff boundary conditions when starting the coupled simulation, mass and heat fluxes from the last 1 kyr of the ice-sheet spin-up are included in the last 5 kyr of ocean spin-up. The initial ice spin-up was done for 200 kyr with PISM v1.0 (similar to Seroussi et al. (2017)) and continued for another 10 kyr with the updated PISM v1.1.4. Ocean temperatures around Antarctica show a warm bias between 0.9 and 3.7 °C, which is too warm to maintain a stable ice sheet when coupled to PISM. Temperature and salinity fields are therefore modified by employing an anomaly method similar to Jourdain et al. (2020). From the ocean fields modeled by MOM5, anomalies relative to the last 100 years of the spin-up are calculated. These anomalies are then applied to the observational input used to drive PICO in the ice sheet spin-up. With this method, the ocean forcing for the ice sheet remains close to the stable forcing as long as the ocean state is not altered.

Starting from the spin-up ice and ocean states, two different coupled experiments are conducted for 4 kyr, both using a 10 year coupling time step. One setup provides the mean ocean forcing over the coupling time step to the ice model, while the

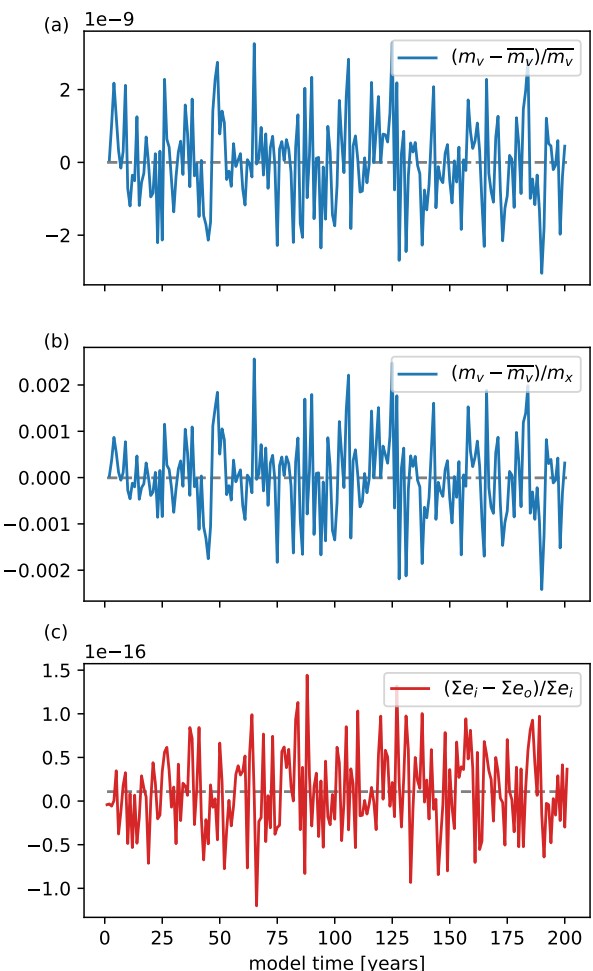

**Figure 9.** Mass and energy conservation. (a) Relative error of virtual mass progression in the coupled ice-ocean system which excludes mass changes applied through surface fluxes and the internal model drift of the coarse grid MOM5/SIS setup. (b) A comparison of virtual mass fluctuations to the mass exchanged between ocean and land ice components ($m_x$). (c) Relative error through remapping energy flux from PISM to MOM5 grid, where $e_i$ and $e_o$ describe the transferred energy fields (unit W) on the land ice and ocean grid respectively. $\Sigma e$ is the spatially aggregated energy over the whole grid domain.

other uses a timeseries forcing of annual averaged ocean temperature and salinity and thus reflects the ocean forcing variability of a yearly coupling time step. Results of both experiments are shown in Fig 10, including the last 5 kyr of standalone spin-ups for comparison. To analyse the ocean state, the following metrics are used: total ocean heat content (Fig. 10a); average of ocean model potential temperatures and salinities in southern most cells at 400 m depth (b,e); Antlantic Meridional Overturning Circulation (AMOC) in panel (c), defined as the maximum annual mean of North Atlantic overturning between 20° N and 90° N and below 500 m; pacific deep temperature (d), which is the ocean potential temperature below 3000 m in the area 110° E –

80° W and 10° S – 70° N; and Antarctic Bottom Water Formation (AABW) in panel (f), which is defined as the maximum annual mean of overturning between 90°S and 0°S and below 2000 m. The state of the Antarctic Ice Sheet is analysed with the following metrics: ice volume above flotation (g); total area of grounded and floating ice (h,i); grounding line movement (j) as the mean of minimum distance between modeled grounding line and Bedmap2 data in every grounding line grid cell; ice thickness evolution (k) as root-mean-squared error (RMSE) of modeled grounded ice thickness compared to Bedmap2 data; and surface velocity deviation (l), defined as RMSE of modeled surface velocities above $100 \, \mathrm{m \, yr^{-1}}$ compared to Ice Velocity Map, v2 (Rignot et al., 2017; Mouginot et al., 2012; Rignot et al., 2011).

The coupled system remains in equilibrium for both scenarios (orange and green lines for ocean; gold and dark grey lines for ice state in Fig. 10) as no major drift can be observed in any of the ocean or ice metrics. Variability of ice volume above flotation (Fig. 10g) is in the range of 0.15 m before and after coupling. The same pattern is observed in total ocean heat content (a) and pacific deep temperature (d), where the latter shows a variability of 0.04 °C. Variations in Antarctic mean ocean temperatures are within 0.1 °C. Changes in AMOC (c) and AABW (f) are in the range of 0.2 and 0.6 Sv, respectively, where $1 \, \mathrm{Sv} = 10^6 \, \mathrm{m^3 \, s^{-1}}$. Variability in the other ice metrics like grounded and floating area (h,i), grounding line deviation (j), ice thickness (k) and surface velocities (l) are comparable between coupled runs and the standalone spin-up. As no significant differences between the two scenarios can be observed, we are concluding that a coupling time step of 10 years is sufficient for coupled experiments that are in equilibrium. Whether this also holds for transient simulations, is yet to be verified.

## 6  Discussion

The framework presented here to couple the ice component PISM to the ocean component MOM5 via PICO fulfills all three goals stated in the Introduction, which are (1) mass and energy conservation across both component domains, (2) an efficient as well as (3) generic and flexible coupling framework design:

As described in Section 5.2, mass conservation across both component domains can be assured. Furthermore, the remapping scheme for energy fluxes is conservative as well. Compared to the required run time of MOM5, the framework routines are very efficient when choosing a coupling time step of 10 years. More frequent coupling causes a larger overhead, as reading and writing the complete model state of PISM to and from files is relatively expensive for very short simulation times. However, an increased ocean to ice forcing of 1 year does not affect the equilibrium state of the coupled system as shown in Section 5.3. The third criterion is fulfilled by the chosen offline coupling approach, which provides a generic and flexible design by making use of the component-related flexibility concerning grid resolution and degree of parallelisation. This does not easily apply to the alternative approach of online coupling, which will be discussed below.

The chosen offline coupling framework executes the two different components alternately and independently and takes care about redistributing the input and output files across the components as explained in Section 3. However, it is also conceivable to adopt an online coupling approach (also called synchronous coupling), where the ice and ocean component code are consolidated into one code structure. The exchange of variables between both components can subsequently take place through access to the same shared memory instead of writing the required variables to disk and reading from there again, as it is done

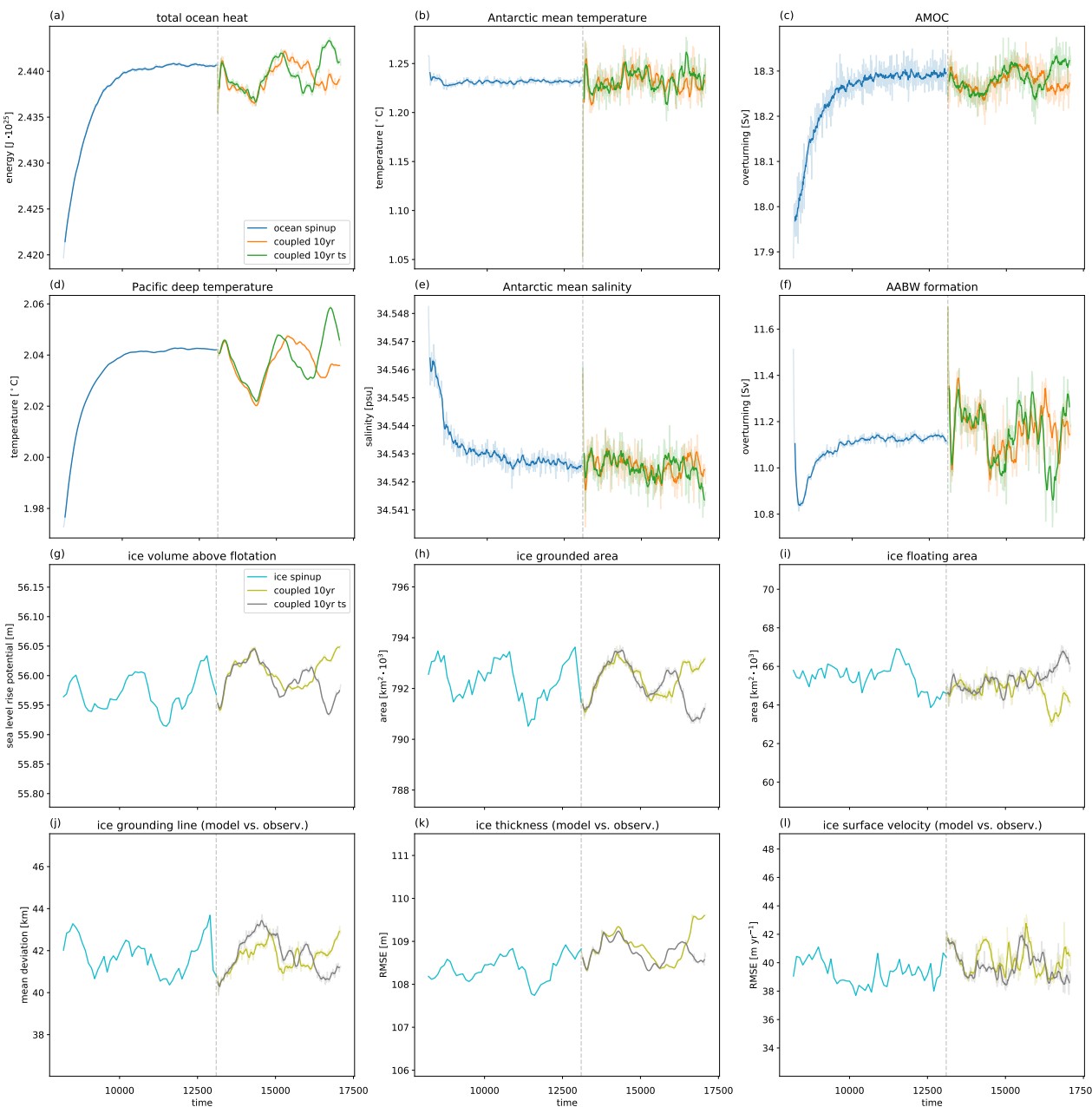

**Figure 10.** Evolution of the Antarctic Ice Sheet and the global ocean during spin-up and coupled simulations under constant climate forcing. Details about ocean (panels a-f) and ice metrics (panels g-l) are given in Section 5.3. Coupling starts at the vertical dashed line. Two coupling variants are presented, both using a coupling time step of 10 years, while one passes the time series of ocean forcing to the ice model (denoted as ts). Light and solid lines are 10 year and 100 year running means, respectively.

in offline coupling. This approach is for instance used by Jordan et al. (2018). A comprehensive framework for online coupling of ocean and ice components is described in Gladstone et al. (2021). This coupling approach is especially powerful for high resolution, cavity resolving ice-ocean coupling, where frequent updates of the ice-shelf cavity geometries and corresponding melt rates are important. However, a prerequisite for online coupling is the adaptation of the standalone models for interactive

execution of subroutines through a defined (external) interface. In the given case of coupling PISM and MOM5, this means that at least one of the two programs' code structure needs major modifications and modularisation to equip the individual component parts like initialisation, time stepping routine, disk I/O, stock checking, etc with suitable interfaces. This is independent of the chosen online coupling design (incorporating one code structure into the other or creating a new master program which governs both components). Synchronisation of the PISM adaptive time step and the fixed ocean component time step would

be a further issue, also keeping in mind that the comparably small ocean time step of a few hours is not applicable for the ice component: PISM can have a time step of around 0.5 years close to equilibrium with $16\,\mathrm{km}$ resolution due to the longer characteristic timescales of ice dynamics. The fact that both components are written in different programming languages (C++ and Fortran) imposes its own (however minor) barriers. A possible benefit of the described online coupling is less disk I/O overhead, which is especially relevant for small coupling time steps in the offline coupling approach (see Sec. 5.1). However,

that does not outweigh the high initial and ongoing development effort which arises through writing and maintaining modified versions of the main component versions. Offline coupling comes with the advantage, that only very minimal modifications of the existing components' source code are necessary. This makes it fairly easy to even replace the ice or ocean components in use with similar existing models, like using MOM5's successor MOM6. A further benefit of the offline coupling approach is that it allows easily to run several independent instances of PISM, e.g. for Antarctica and Greenland, at the same time.

The coupling implementation exhibits certain simplifications that can be subject of future improvements. As described in Section 4.2, the mass and energy fluxes computed from PISM output are given as input to the ocean surface rather than being distributed throughout the water column - a limitation of MOM5's simplified treatment of all land-derived mass fluxes, including those from ice sheets. This simplification may affect vertical heat distribution and local sea ice formation (Bronselaer et al., 2018) as near-surface input generally makes the vertical column more stratified, whereas input below the mixed layer

destabilises the water column, thus enhances vertical mixing and extends the mixed layer depth (Pauling et al., 2016). A more realistic input depth into the ocean would be the lower edge of the ice shelf front (see start of upper green arrow in Fig. 3; Garabato et al., 2017) which could be determined as the average ice-shelf depth of the last PICO box.

      Mass and energy fluxes are composed of basal melting, surface runoff and calving and provided as input to the southernmost ocean cells (see Sec. 4.2). Icebergs can however travel substantial distances before they have been melted completely and thus

continuously distribute mass and energy fluxes into the ocean (Tournadre et al., 2016). The resulting spatial distribution of iceberg fluxes can introduce biases in sea-ice formation, ocean temperatures, and salinities around Antarctica (Stern et al., 2016). Currently this is not considered in our framework and may be simulated by an additional iceberg component (like described in Martin and Adcroft, 2010) in the future.

      Another simplification is contained in the energy flux description from ice to ocean. As explained in Section 4.2, the flux is

calculated as the energy transferred through phase change from frozen ice to liquid water. Diffusion of heat through the ice and

energy required to warm up melt water to ambient ocean temperatures are currently not considered as they are estimated to be comparably small (Holland and Jenkins, 1999).

The waxing and waning of ice sheets on glacial-interglacial time scales causes transfer of large amounts of water between the oceans and land ice sheets. Significant changes in sea level (120-135 meters below present during the last glacial maximum (Clark and Mix, 2002)) have large impacts on coast line positions. The response of the solid Earth component to changes of ice-sheet mass has a similar effect. During long simulations the land-ocean mask needs to be adapted accordingly. As MOM5 cannot handle mixed ocean-land cells, which would allow for a smooth adaption of a changing coast line, major changes in the land-ocean mask need to be performed during a transient simulation. This requires careful considerations like the initialisation of newly flooded cells and implications concerning mass and energy conservation as well as model stability. The development of a sea-level based dynamic ocean domain adaptation which applies the described changes to new ocean restart conditions is currently under way and will be incorporated in the described coupled setup in the future.

In this study we focus on the technical implementation of the coupling framework and evaluate it in a transient simulation under constant present-day climate forcing. As the ocean component has warm biases at intermediate depth around the Antarctic margin, we apply an anomaly approach to avoid unrealistic high melting and obtain physically meaningful simulations of the coupled system. We add anomalies from the ocean model component to observed temperatures, similar to the approach in ISMIP6 (Jourdain et al., 2020; Nowicki et al., 2020). The difficulties to accurately simulate Antarctic shelf dynamics and deep water formation in the Southern Ocean with ocean general circulation models is a long standing issue for the ocean modelling community, with almost no models of the CMIP5 generation able to do this successfully (Heuzé et al., 2013). The improvement of these biases is the subject of ongoing work via the implementation and tuning of the new MOM6 ocean model. While the anomaly approach is appropriate for present-day simulations, for which we have observations, it is as yet unclear how these biases might be addressed for transient simulations on multi-millennial time-scales. In the transient simulations the effect of using a 10-yearly coupling time step was tested in a simulation with the variable 10-year ocean forcing being applied to the ice sheet instead of the 10-year average. We find that this variability has no effect in a steady-state simulation. These open issues, including the choice of the coupling time step under physical aspects, will be considered in a future study.

The presented coupling framework is characterized by a reduced complexity approach. This is reflected for instance in the basin wide averaging of PICO input which does not account for horizontal differences such as cavity in- and outflow regions or modification of water masses on the continental shelf. Similarly, the complex processes determining whether upwelling Antarctic Circumpolar Deep Water reaches the continental shelf and the grounding lines (Nakayama et al., 2018), can only be partly represented due to the coarse bathymetric features of the MOM5 grid (see also Fig. 1b). However, the intermediate complexity of the coupled system enables ocean simulations on a global domain, opening possibilities to study interactions, feedbacks and possible tipping behaviour on millennial time scales. Overall, despite the limitations discussed above, the coarse grid setup of MOM5 in combination with the representation of the ice pump mechanism in PICO, makes large-scale and long-term ice-ocean coupling possible at an intermediate level of complexity.

## 7 Conclusions

In this study we focus on the technical approach and conservation aspects of coupling a large-scale configuration of the ice sheet model PISM and a coarse grid resolution setup of the ocean model MOM5 via the cavity module PICO. This allows to capture the typical overturning circulation in ice-shelf cavities that cannot be modeled in large-scale ocean standalone models. We can assure that conservation of mass and energy is obtained in the coupler between the ocean and land ice components while having a computationally efficient and flexible coupling setup. Using this framework, which is openly available and can also be transferred to other ice-sheet and ocean general circulation model components, feedbacks between the ice and ocean can be analysed in large-scale or long-term modelling studies. In future work, the physical processes and feedbacks between ice-sheet, ice shelves and ocean will be further analysed and the interaction strengths can be evaluated on various timescales, from decades to multi-millennial simulations.

*Code and data availability.* The coupling framework code is hosted at https://github.com/m-kreuzer/PISM-MOM_coupling (last accessed: April 16, 2021). The exact version used in this paper has been tagged in the repository as v1.0.3 and is archived via Zenodo (https://doi.org/10.5281/zenodo.4692679, last accessed: April 16, 2021). The code makes use of the software tools Climate Data Operator (CDO, version 1.9.6, Schulzweida (2019); https://doi.org/10.5281/zenodo.3991595, last accessed: April 16, 2021) as well as the netCDF Operator (NCO, version 4.7.8, Zender (2018); https://doi.org/10.5281/zenodo.1490166, last accessed: April 16, 2021). The Parallel Ice Sheet Model (PISM) was used in version 1.1.4 (https://doi.org/10.5281/zenodo.4686967, last accessed: April 16, 2021) and the Modular Ocean Model (MOM) was used in version 5.1.0 with slight modifications as archived in https://doi.org/10.5281/zenodo.3991665 (last accessed: April 16, 2021). All data used in the tests above is archived in https://doi.org/10.5281/zenodo.4692940 (last accessed: April 16, 2021).

## Appendix A: Benchmark results

*Author contributions.* MK wrote and implemented the coupling framework and performed the analysis. RW, GF and SP conceived the study. MK and RR designed the coupling strategy via PICO. SP and WH provided support with the setup and use of MOM5. RR and TA provided support with the use of PISM. RR contributed to shaping the manuscript. MK prepared the manuscript with input and feedback from all co-authors.

*Competing interests.* The authors declare that they have no conflict of interest.

*Acknowledgements.* Development of PISM is supported by NASA grant NNX17AG65G and NSF grants PLR-1603799 and PLR-1644277. The authors gratefully acknowledge the European Regional Development Fund (ERDF), the German Federal Ministry of Education and Re-

**Table A1.** Runtimes of coupled PISM-MOM5 setup for 200 years model time using 32 cores. PISM runtimes include PICO and MOM5 runtimes include SIS and FMS components.

| routine | 1 year coupling | | 10 year coupling | |
|---|---|---|---|---|
| | time [s] | ratio [%] | time [s] | ratio [%] |
| total | 21976.49 | 100.00 | 13244.80 | 100.00 |
| preruns | 24.17 | 0.11 | 24.41 | 0.18 |
| preprocessing | 40.97 | 0.19 | 43.03 | 0.32 |
| MOM runs | 15446.26 | 70.29 | 12267.26 | 92.62 |
| MOM postprocessing | 1993.09 | 9.07 | 205.98 | 1.56 |
| PISM runs | 2830.57 | 12.88 | 467.26 | 3.53 |
| MOM-to-PISM processing | 861.89 | 3.92 | 125.43 | 0.95 |
| PISM-to-MOM processing | 90.43 | 0.41 | 14.01 | 0.11 |
| concatenating output files | 656.44 | 2.99 | 81.91 | 0.62 |

search and the Land Brandenburg for supporting this project by providing resources on the high performance computer system at the Potsdam Institute for Climate Impact Research. M.K. is supported by the Deutsche Forschungsgemeinschaft (DFG) by grant WI 4556/4-1. R.R. was supported by the Deutsche Forschungsgemeinschaft (DFG) by grant WI 4556/3-1 and through the TiPACCs project that receives funding from the European Union's Horizon 2020 research and innovation programme under grant agreement no. 820575. T.A. is supported by the Deutsche Forschungsgemeinschaft (DFG) in the framework of the priority program "Antarctic Research with comparative investigations in

Arctic ice areas" by grant WI 4556/2-1. The work by T.A., R.W. (FKZ: 01LP1925D) and W.H. (FKZ: 01LP1504D and FKZ: 01LP1502C) has been conducted in the framework of the PalMod project, supported by the German Federal Ministry of Education and Research (BMBF) as Research for Sustainability initiative (FONA).

We thank Paul Gierz from AWI for in-depth discussions in the initial phase of this project. Significant parts of the work were done while M.K. was affiliated to University of Potsdam, Department of Computer Science, August-Bebel-Str. 89, 14482 Potsdam, Germany. Many

thanks to Prof. Dr.-Ing. Christian Hammer for supervision of M. K.'s Master Thesis with the title "Coupling the Ice-Sheet Model PISM to the Climate Model POEM" which laid the foundation for this publication.

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
