# Peer review of "Coupling framework (1.0) for the ice sheet model PISM (1.1.4) and the ocean model MOM5 (5.1.0) via the ice-shelf cavity module PICO in an Antarctic domain"

_Geoscientific Model Development, 2020_

## Referee Comment (RC1) · Anonymous Referee #1 · 8 Oct 2020

"Coupling framework (1.0) for the ice sheet model PISM (1.1.1) and the ocean model MOM5 (5.1.0) via the ice-shelf cavity module PICO" by Kreuzer et al. describes the software implementation of an ice-sheet/ocean coupler, designed for ocean models that do not resolve ice shelf cavities. This manuscript details the algorithm by which data is exchanged between the two models on a basin-averaged basis, while ensuring conservation and a reasonable computational overhead. The scientific validation of the coupled system is left for another paper, so the realism of the model is not discussed.

General comments:

Coupling ocean and ice-sheet models via PICO is a worthwhile idea, as it would fill a

gap in coupled modelling. If successful, it would enable CMIP-style models to include evolving ice sheets without having to explicitly resolve ice shelf cavities, which would open a lot of doors for long-timescale simulations. The coupling algorithm described here is well explained and logically designed. I have a few questions about the coupler which should be addressed for clarity (see my specific comments below).

However, I am concerned at how this paper does not show whether the coupled system can actually produce reasonable results, which are acceptably realistic and stable for the present-day climate. I find it acceptable to separate the software description from the model validation, but only if the model validation is submitted as a companion paper at the same time. Without this assurance, I worry that the authors have not yet tested the realism of the coupled system.

What if there are insurmountable challenges which make the premise of the coupling unusable? For example, Figure 5 suggests that the ocean temperatures being passed to PICO are far too warm, with no continental shelf temperatures below about 0.25C. In reality, we know that large regions of Antarctica have inflow into ice shelf cavities consistently around -1.9C, as a result of sea ice formation. Effectively, coupling with this MOM configuration will turn all ice shelf cavities into warm cavities like the Amundsen Sea, but we know that many/most cavities are cold cavities. Surely this would lead to unacceptably high basal melt rates and grounding line retreat.

What if these challenges can only be overcome by substantially changing the coupling design, at which point this paper becomes out of date? While I applaud the authors for their worthwhile efforts in this model development, I worry it is premature to publish before we know whether this approach will work. It might be best to save this paper up until the scientific validation has been completed and written up. At that point, it will be a very nice submission to GMD.

My other major comment is regarding the introduction. I feel that a stronger and clearer case could be made for why this coupling advance is needed. I would love to see a

revised and reorganised introduction, which more clearly lists the major categories of models which already exist (global coupled models with fixed ice sheets; standalone ice sheet models with highly simplified ocean forcing; high-resolution ocean models with static or dynamic ice shelf cavities) and why they are not suitable for ice-sheet/ocean coupling on millennial timescales. This would make it more clear to the non-expert reader why PICO coupling is a major advance.

Specific comments:

Title: Is it necessary to include the version numbers for PISM and MOM5, as well as the coupler? This detracts from readability, and it seems like the coupling should be more or less independent of the specific model versions. Also, the title should specify that the coupling is specific to the Antarctic Ice Sheet.

Abstract, line 8: "Earth system" should be changed to "ocean", since this manuscript doesn't discuss coupling between the ice sheet and other climate model components.

Line 24: The discussion of ocean forcing in paleoclimate should be expanded, as many of the readers may be unfamiliar with this.

Line 26: "prescribed ice-sheet configurations" is not clear - does this mean there is a fixed ice surface topography and no simulation of ice dynamics?

Line 30: I suggest changing "circulation in ice-shelf cavities" to "ice pump", as PICO doesn't explicitly simulate cavity circulation either.

Line 38: Note that Scenario A1B is from CMIP3, not CMIP5/6.

Line 40: It is not clear that "ice-sheet/ocean interactions" here refers to fully coupled ice-sheet/ocean models including ice dynamics. There are many ocean models which simulate ice shelf thermodynamics alone, with static cavities, which is an important distinction from the types of models I think you are referring to.

Line 46: Donat-Magnin et al. (2017) is not a coupled ice sheet-ocean model, but rather

has static cavities as above, and prescribed a different grounding line for a sensitivity experiment. I would suggest just taking this reference out.

Line 60: Surely it is only 2-dimensional ice sheet models, with SSA or similar? I equate 3-dimensional with full Stokes, which I don't think PICO has been coupled to.

Line 60: Change "approximates" to "parameterises"

Line 100: I suspect "poles" and "equator" are the wrong way round - usually meridional resolution is finer at the poles, to follow the convergence of zonal resolution. Coarser meridional resolution at the poles, as the text says, would lead to a strange cell aspect ratio.

Line 101: Is p* the same as z* (which is a more common vernacular for ocean modellers)?

Line 102: What is the vertical resolution of the thickest layers?

Line 113: The text describes the models as run in alternating order. Is it possible to run them simultaneously on different processors? This would save on walltime, although the CPU hours wouldn't change.

Line 114: Figure 4 suggests that MOM is the first model to run. Is this always the case, or can PISM start the chain?

Figure 4: It might be helpful to align the time axes. It's difficult to tell which simulation segments are supposed to coincide with which.

Line 147: Why not extrapolate the values from the centres to the cell boundaries where needed, to minimise the regions of missing data? Would this change anything, or is it implicit in your next point about filling the missing regions?

Line 148: It is unclear whether the missing values are filled as a single block of adjacent missing values, or cell by cell. This would lead to slightly different behaviour in the averaging routine.

Line 154: What happens to the vertical interpolation if the ocean bathymetry is shallower than the continental shelf depth as seen by PISM?

Line 163: If I understand correctly, every coastal ocean cell in a given basin receives the same mass of freshwater from PISM (and similarly for energy). Why is this not scaled by area, so that larger ocean cells receive more freshwater? This would be equivalent to distributing the total mass flux evenly over the coastline in the given region, regardless of the details of the ocean grid.

Figure 7: It's confusing to compare the different units in (a) and (b). Over what time interval are the fluxes in (a) integrated?

Line 171: What happens if the energy flux causes supercooling of the ocean waters? Is this supercooling automatically removed by the sea ice model?

Line 175: Doesn't ignoring these heat fluxes mean the coupled model does not conserve energy? I understand the argument later that the coupler itself conserves energy, during the regridding process, but it appears false to claim that the entire system is conservative (as is implied in line 317).

Line 188: Explain why the coupling time step is an important parameter. What do you expect would happen if it were too long or too short?

Line 200: Clarify that MOM5 runtimes are slightly longer due to a greater fraction of time spent initialising, as with PISM (I assume this is the case).

Line 222: Does this statement mean that MOM5 does not conserve mass? That is concerning, and should be explained further and referenced.

Line 250: As the Galton-Fenzi reference only refers to an EGU presentation, a better reference for online coupling would be Jordan et al. 2018 (doi:10.1002/2017JC013251) which is already published in JGR.

Line 257: I'm not sure it's essential for online coupling to have the same timestep for ice

and ocean, as ocean models often subcycle different timesteps for different processes (eg barotropic vs baroclinic modes).

Line 273: The ocean mixed layer will rarely extend as deep as the ice shelf front. Ice shelf meltwater entering the ocean at depth would therefore destabilise the water column and deepen the mixed layer, whereas applying this flux at the surface would have the opposite effect. I appreciate that applying meltwater at depth is not trivial in MOM5, but more attention should be given to the possible negative impacts of this design choice on the ocean simulation.

————————————————

---

## Referee Comment (RC2) · Rupert Gladstone (Referee) · 7 Jan 2021

The paper describes an implemented approach to interactive coupling between an ice sheet model and an ocean model through a reduced complexity ocean cavity model. The project provides a compromise between complexity and efficiency, allowing large scale coupled simulations in which at least some aspects of cavity circulation are represented. This is a useful contribution to global modelling efforts. For the most part, the work is very clearly described and presented. The figures are well prepared and appropriate. I would recommend publication of this paper with some fairly minor modifications.
**General comments**

I'm not entirely sure what other models might be competing with the current study in terms of global coupled models of intermediate complexity, but I think maybe the UKESM falls into this category? Are there publications about UKESM and how does it compare to your approach? Perhaps it is significantly more computationally expensive than your setup? I think it uses BISICLES and NEMO for ice and ocean models.

The first reviewer recommended including model validation in this paper. Robust model validation is a very large challenge and is not typically included in model description papers. I would prefer to see separate studies present some level of validation rather than try to include it here. Note that this is a highly tunable model, and so a good match to observations over a short time period should be straight forward to achieve. The real challenge will be in quantifying the uncertainty in the model as conditions evolve significantly over long periods of time, and this challenge is well beyond the scope of the current paper. The paper does present some level of model verification, especially regarding conservation, and this is useful.

I am aware that some aspects of my review read like an advertisement for the Earth System Modelling Framework (ESMF)! I shoud clarify I am fully independent of ESMF; I'm just an end user. In my experience ESMF is a very robust, user-friendly, well-documented code. So I guess I'm just a fan! It is a valid point though. Existing coupling frameworks (and I talk about ESMF rather than OASIS or others simply because I have the most experience with it) do provide more features and flexibility than the authors convey knowledge of in some places (see also my specific points below).

It is very interesting to learn that the PISM intialisation time becomes a significant factor as coupling timestep is shortened. It makes the question "what coupling timestep do you need¿' rather important. I couldn't find a comparison of important result metrics between the 1 year and 10 year coupling timesteps. Information about computational cost is given, but is the actual behaviour different? For example, do they give the same

total melt rate over the 200 years? Does the ice shelf thickness evolution look the same in both simulations? And grounding line and ice front evolution? You can't really address whether your framework meets requirement 2 until you've established whether or not a 10 year coupling timestep is sufficient.

I didn't find information about parallelism of the coupling. I infer from this lack of information that the coupling (bash script and file manipulation) all occurs on one processor. If this is the case, can you comment on how many processors you intend to use for production runs, and whether this might become a bottle neck for larger parallel simulations? For example, the ocean postprocessing and intermodel processing took 15

The Earth System Modelling Framework (ESMF) community adopts some terminology that I find quite beneficial in that it offers clarity in certain areas that can otherwise become slightly confusing. Individual models in a coupled system are referred to as components. This avoids confusion between the coupled model and the individual ice and ocean models. I like this clarity and I think it would be nice the the authors adopt it, but I do not wish to make this a requirement, just a suggestion.

**Specific comments**

Line 107: I don't quite understand the use of two horizontal dimensions. If I understood right, PICO just represents the overturning circulation. How do two horizontal dimensions come into play here? Or perhaps I should go and read Ronja's 2018 paper on PICO (lazy reviewer!)!

Lines 113-114: This is also known as "sequential coupling".

Line 115: In this context, does an "integration step" mean running the model for a coupling time step? Can you clarify this in the text?

Line 118: "the last of these averaged fields" is a slightly confusing expression. Presumably PISM receives the average of the fields over the full (10 years in this example)

coupling timestep? But this isn't clear to me from the chosen wording.

Figure 4: There appears to be a temporal offset between $t_{ice}$ and $t_{ocean}$ in the way that the Figure is presented. But the text suggests that the time period over which the two components are integrated should not be offset (line 113 refers to the "same model time"). Can you clarify this? It is not clear to me what is meant be "Sharing the same time axis", perhaps this is related to my question?

Line 134: I don't know what "entanglement" means in this context.

Line 139: Is "surrounding ice sheets" a typo in this context? I mean, the Zwally basins divide up the Antarctic Ice Sheet... perhaps you mean shelves not sheets here?

Lines 148-149: If you use only adjacent cells to populate missing values, presumably you iterate until all cells have a value? I mean there must, to start with, be plenty of cells that are not adjacent to a cell with a value.

Lines 159-160: I'm fairly confident that ESMF's "common" regridding algorithms include masked nearest neighbour remapping options that are very simliar to what is described here. I intend to use these for remapping subglacial outflow from an ice model to an ocean model, though I haven't actually implemented this yet, and it looks like the required functionality to do this in a mass conserving way is already in place.

Section 5.1. I can't quite make the numbers add up here. The run with a 1-year coupling timestep takes 22700s. The ocean post-processing, interprocessing and PISM percentages given add up to 35

Line 193: I presume "total runtime" is the elapsed time not the cpu hours? i.e. the total computational time would be this number times 32?

Equation 1: Surely mass has dimensions of mass and smb has dimensions of mass/time. So how can you simply add these? I don't understand why there would be rates in this equation. Surely the total mass is the sum of the mass from each component at any given moment in time? The same comment for lines 214-215. How can

you subtract a flux from a mass? I can't make sense of it!

Line 222: It is not clear from this description whether the dimensions of $d_{osi}$ should be mass or mass/time.

Figure 9: To put this apparently small error into context, it would be useful to give some indication of how much mass is transferred between ice and ocean. I think the total mass transfer is probably a more relevant figure here than the total ice or ocean mass. We need to know that the error measures are small not only compared to the total mass of the coupled system, but also small compared to the amount of mass being transferred between ice and ocean over the intergration period.

Line 239: I don't think you can make this unqualified statement that the framework fulfills all three requirements. 1 and 3, yes, sure, but requirement 2 is really only partially fulfilled, as you continue on to discuss in the following paragraph.

Paragraph 248-267. This paragraph completely omits to discuss alternative forms of online coupling in which a single executable links to component runtime libraries and fulfills a role equivalent to your bash script as a master (or parent) program. The implication in this paragraph is that one must choose between an offline coupling in which components are called independently and an online coupling in which one component must be the master and the other the slave. But this is not the choice that a coupled model developer faces, there are many more options available. For example, the Earth System Modelling Framework (ESMF) offers full flexibility in this sense. The developer can choose whether to create a new parent routine or to establish one component as master. You might want to cite Gladstone et al GMDD FISOC paper (published in June 2020 as a discussions paper and now accepted pending minor revisions for GMD), which is essentially an online equivalent of your coupling structure, with a new (Fortran in this case) master program that calls the child components (which have been made ESMF compatible and compiled as libraries). This approach also allows flexibility in terms of switching between components (indeed we currently have a choice of two

ocean models coupled through FISOC, and two furtehr ice sheet models are in the process of being incorporated). Of course there is still some overhead in terms of ensuring each component is compatible with the coupling framework (ESMF in the case of FISOC), so it is a longer development path than your bash script plus file manipulations, but perhaps not as onerous or restrictive as implied in the current version of the text.

Line 256. ESMF (for example) handles all parallel regridding in an efficient manner. So long as component mesh and field information can be made available in ESMF runtime data structures (which I acknowledge requires some coding and may not be trivial), the regridding between different partitioning is all handled automatically.

Line 260. The C/Fortran issue is a fairly small technical issue. There are plenty of codes around that use both at runtime.

Lines 269 – 274. The physical implications of this issue could be fairly interesting. Could the input ever come in beneath the turbulent mixed layer? Could the input of fresh water at the surface have a stabilising effect that would not occur if it was mixing up from lower down? Do you have any plans to investigate this further? This could all raise interesting questions that have been brushed over very lightly here.

Lines 285 – 288. This is actually a very important issue, especially because you intend this framework to be applied to long timescale simulations. You mention that work is in progress, but can you say a few lines about how this will be implemented? Can you also ive an example or two of the how you intend to use the model in its current form so that the reader can start to envisage how much of an issue (or not) the lack of evolving active ocean domain is?

Lines 293-295. These lines seem to imply that tuning PICO will somehow make up for the lack of a full representation of the complex 3D ocean circulation over the continental shelf and under ice shelves. This will not, in general, be the case. It is clear that this model is a compromise approach, a model of intermediate complexity with the benefit

of efficiency. This has value; you don't need to try too hard to defend or justify limitations of PICO. I would prefer to see the limitations presented directly without implying that they can be overcome (through tuning for example).

311. "appropriate" is not well defined in this context, and perhaps not yet fully justified. How about just "intermediate" instead?

––––––––––––––––––––––––––––––––

---

## Author Comment (AC1) · 4 Mar 2021

**Author comment on RC1 and RC2 of `gmd-2020-230`**

Moritz Kreuzer        Ronja Reese        Willem Nicholas Huiskamp        Stefan Petri
Torsten Albrecht        Georg Feulner        Ricarda Winkelmann

March 4, 2021

This statement is addressing the reviews of the discussion paper "*Coupling framework (1.0) for the ice sheet model PISM (1.1.1) and the ocean model MOM5 (5.1.0) via the ice-shelf cavity module PICO*" (`gmd-2020-230`) in Geoscientific Model Development.

We thank the editor Steven J. Phipps, the first anonymous reviewer and the second reviewer Rupert Gladstone for their interest and time reviewing our manuscript and their thoughtful, constructive and very helpful comments and feedback. We provide detailed responses to all comments below, where the reviewers remarks are displayed as indented quotes. Changes in the manuscript (italics) are given where appropriate, showing old text in red and new text in blue color.

**Comments by Reviewer 1**

> "Coupling framework (1.0) for the ice sheet model PISM (1.1.1) and the ocean model MOM5 (5.1.0) via the ice-shelf cavity module PICO" by Kreuzer et al. describes the software implementation of an ice-sheet/ocean coupler, designed for ocean models that do not resolve ice shelf cavities. This manuscript details the algorithm by which data is exchanged between the two models on a basin-averaged basis, while ensuring conservation and a reasonable computational overhead. The scientific validation of the coupled system is left for another paper, so the realism of the model is not discussed.

Thanks for the review of our manuscript. To address the major comments of the first reviewer, we will implement the following changes in the revised version of the manuscript:

- We will add results of multi-millennial coupled runs that demonstrate that a coupled configuration for present-day conditions remains stable.

- We will revise the introduction to include a more comprehensive discussion of already existing coupled models and approaches as well as the novelty of the approach presented in our manuscript.

**General Comments**

Coupling ocean and ice-sheet models via PICO is a worthwhile idea, as it would fill a gap in coupled modelling. If successful, it would enable CMIP-style models to include evolving ice sheets without having to explicitly resolve ice shelf cavities, which would open a lot of doors for long-timescale simulations. The coupling algorithm described here is well explained and logically designed. I have a few questions about the coupler which should be addressed for clarity (see my specific comments below).

However I am concerned at how this paper does not show whether the coupled system can actually produce reasonable results, which are acceptably realistic and stable for the present-day climate. I find it acceptable to separate the software description from the model validation, but only if the model validation is submitted as a companion paper at the same time. Without this assurance, I worry that the authors have not yet tested the realism of the coupled system.
What if there are insurmountable challenges which make the premise of the coupling unusable? For example, Figure 5 suggests that the ocean temperatures being passed to PICO are far too warm, with no continental shelf temperatures below about 0.25C. In reality, we know that large regions of Antarctica have inflow into ice shelf cavities consistently around -1.9C, as a result of sea ice formation. Effectively, coupling with this MOM configuration will turn all ice shelf cavities into warm cavities like the Amundsen Sea, but we know that many/most cavities are cold cavities. Surely this would lead to unacceptably high basal melt rates and grounding line retreat.
What if these challenges can only be overcome by substantially changing the coupling design, at which point this paper becomes out of date? While I applaud the authors for their worthwhile efforts in this model development, I worry it is premature to publish before we know whether this approach will work. It might be best to save this paper up until the scientific validation has been completed and written up. At that point, it will be a very nice submission to GMD.

Thanks for addressing the very important point of model validation. We agree that the coupling framework can only be useful if it is able to reproduce a stable coupled setup for the present-day climate. In the originally submitted manuscript (lines 296-307), we suggested a bias correction of the modelled ocean properties with respect to present-day observations as used for example in the ISMIP6 project (Jourdain et al., 2020). We here follow this approach in coupled simulations. For validation of the framework, we will include coupled model results in the revised manuscript that show the framework's ability to represent the present-day climate and ice sheet state in a stable manner over millenial time scales. We want to note that the warm bias in ocean temperatures in our model can probably be attributed to the coarse model resolution, however, such bias has also been found in CMIP5 models with higher resolution (Heuzé et al., 2013; Barthel et al., 2020).

My other major comment is regarding the introduction. I feel that a stronger and clearer case could be made for why this coupling advance is needed. I would love to see a revised and reorganised introduction, which more clearly lists the major categories of models which already exist (global coupled models with fixed ice sheets; standalone ice sheet models with highly simplified ocean forcing; high-resolution ocean models with static or dynamic ice shelf cavities) and why they are not suitable for ice-sheet/ocean coupling on millennial timescales. This would make it more clear to the non-expert reader why PICO coupling is a major advance.

We thank the reviewer for this constructive comment and agree that the introduction could be improved with a more comprehensive discussion of existing coupling approaches including the novelty of our approach using PICO. We will incorporate the reviewer's suggestions when submitting a revised manuscript including the other changes (see comments below).

**Specific Comments**

> Title: Is it necessary to include the version numbers for PISM and MOM5, as well as the coupler? This detracts from readability, and it seems like the coupling should be more or less independent of the specific model versions. Also, the title should specify that the coupling is specific to the Antarctic Ice Sheet.

The title includes the version numbers to meet the requirements of GMD model description papers[1]. To reflect the Antarctic application of the described coupling, we will change the title of the finalised manuscript to *"Coupling framework (1.0) for the ice sheet model PISM (1.1.1) and the ocean model MOM5 (5.1.0) via the ice-shelf cavity module PICO in an Antarctic domain"*.

> Abstract, line 8: "Earth system" should be changed to "ocean", since this manuscript doesn't discuss coupling between the ice sheet and other climate model components.

We agree with the reviewer that the suggested wording is more precise. The text has been changed accordingly.

> Line 24: The discussion of ocean forcing in paleoclimate should be expanded, as many of the readers may be unfamiliar with this.

Thanks for making this helpful suggestion. We will incorporate this in the revised introduction (see above).

> Line 26: "prescribed ice-sheet configurations" is not clear - does this mean there is a fixed ice surface topography and no simulation of ice dynamics?

The reviewer is correct in their interpretation. We apologise that this was not sufficiently clear. The passage has been rephrased as follows:

*While ice-sheet simulations usually rely on external forcing without interactive coupling to calculate sub-shelf melt rates (Pollard et al., 2016; Sutter et al., 2016; Albrecht et al., 2020), general circulation models usually use prescribed ice-sheet configurations with fixed ice-surface topography and no simulation of ice dynamics (Kageyama et al., in review, 2020).*

> Line 30: I suggest changing "circulation in ice-shelf cavities" to "ice pump", as PICO doesn't explicitly simulate cavity circulation either.

We agree that this is more appropriate and have changed the manuscript accordingly:
* * *
[1]https://www.geoscientific-model-development.net/about/manuscript_types.html#item1

*These coupled setups are using, if they consider ocean forcing on the ice sheet, simple melt parameterisations that do not take the*  *ice pump into account, which is important to estimate realistic melt rates.*

> Line 38: Note that Scenario A1B is from CMIP3, not CMIP5/6.

Thanks for pointing this out. As the specific scenario is not of relevance here, it might be best to omit this information here. We will consider this in the revised introduction (see above).

> Line 40: It is not clear that "ice-sheet/ocean interactions" here refers to fully coupled ice-sheet/ocean models including ice dynamics. There are many ocean models which simulate ice shelf thermodynamics alone, with static cavities, which is an important distinction from the types of models I think you are referring to.

We agree that the chosen wording leaves room for interpretation and thank the reviewer for addressing this. This is in line with the reviewer's major comment about reorganising the introduction with a more detailed discussion of existing ice-ocean coupling models and approaches. We will therefore take this remark into account in the revised version of the introduction before resubmitting the manuscript.

> Line 46: Donat-Magnin et al. (2017) is not a coupled ice sheet-ocean model, but rather has static cavities as above, and prescribed a different grounding line for a sensitivity experiment. I would suggest just taking this reference out.

We thank the reviewer for spotting this error. The reference has been removed as suggested.

> Line 60: Surely it is only 2-dimensional ice sheet models, with SSA or similar? I equate 3-dimensional with full Stokes, which I don't think PICO has been coupled to.

We agree that SSA models could be called 2D, however, PISM is a hybrid model that employs a superposition of the SSA and SIA. Since the SIA is vertically non-uniform, PISM is a 3D model with a 3-dimensional field of ice velocities. It also solves thermodynamics on a 3-dimensional grid. Because the term '2-dimensional' could be misunderstood as flowline models with one horizontal and one vertical dimension, we have changed the wording in the manuscript as follows:

*PICO extends the ocean box model by Olbers and Hellmer (2010) for application in*  *ice sheet models that resolve both horizontal dimensions.*

> Line 60: Change "approximates" to "parameterises"

This has been changed as requested.

> Line 100: I suspect "poles" and "equator" are the wrong way round - usually meridional resolution is finer at the poles, to follow the convergence of zonal resolution. Coarser meridional resolution at the poles, as the text says, would lead to a strange cell aspect ratio.

The description in the manuscript is correct, see also the model description paper from Galbraith et al. (2011, Appendix A.b). For illustration we have added a figure of the meridional resolution of the MOM5 ocean grid in use, see Figure 1. A different point (which hasn't explicitly mentioned so far) is that the grid is of tripolar structure to avoid a singularity at the north pole. This information has been added to the manuscript:

*For this study, MOM5 is used with a global coarse grid setup from Galbraith et al. (2011, see Fig. 1b): the lateral model grid is 3° resolution in longitude (120 cells) and in latitude it varies from 3° at the poles to 0.6° at the equator (80 cells). It makes use of a tripolar structure to avoid the grid singularity at the North Pole (Murray, 1996).*

[Figure]

Figure 1: Meridional resolution of MOM5 ocean grid.
* * *
Line 101: Is p* the same as z* (which is a more common vernacular for ocean modellers)?

$p^*$ and $z^*$ are not synonymous, but similar coordinate systems. While $p^*$ is a pressure-based coordinate extending over the time-independent range $0 \leq p^* \leq p_{bottom}$, $z^*$ is similarly defined in depth space, spanning $-H \leq z^* \leq 0$ where $-H$ equals the local ocean depth. More information about the MOM5 internals can be found in the manual[2].
* * *
Line 102: What is the vertical resolution of the thickest layers?

As the vertical resolution increases with depth, the lowermost vertical layer is the thickest. It has a maximum resolution of $\sim 513$ dbars which translates to a thickness of $511\,\mathrm{m}$ when using hydrostatic water pressure conversion with density $\rho = 1023.6$ kg/m$^3$ and $g = 9.80665$ m/s$^2$. As described in Galbraith et al. (2011, Appendix A.b), the model employs partial bottom cells to allow a more realistic representation of the bathymetry. The vertical resolution of the MOM5 grid in use is shown in Figure 2. To make this more clear in the manuscript, the text has been extended:

*The vertical grid is defined using the re-scaled pressure coordinate ($p^*$) with a maximum of 28 vertical layers. The uppermost eight layers are approximately 10m thick, gradually increasing for deeper cells to a maximum of ca. 511m. The vertical resolution in depths relevant for ice shelf cavities is between 50 and 180m. The lowermost cells can have a reduced thickness to account for ocean bathymetry with partial cells.*
* * *
[2]https://mom-ocean.github.io/assets/pdfs/MOM5_manual.pdf, last accessed: 4th March 2021

[Figure]

Figure 2: vertical resolution of MOM5 ocean grid

> Line 113: The text describes the models as run in alternating order. Is it possible to run them simultaneously on different processors? This would save on walltime, although the CPU hours wouldn't change.

Simultaneous runs introduce a lag of one coupling time step in the exchange of variables between the models. While this can be acceptable, a concurrent setup makes most sense, when both models require roughly the same computation time. Otherwise one set of processors will be idle most of the time which is bad from an efficiency perspective. In the 10 year coupling setup, the major share of runtime is consumed by the ocean model (see Section 5.1) and a concurrent setup would hardly reduce the runtime.

> Line 114: Figure 4 suggests that MOM is the first model to run. Is this always the case, or can PISM start the chain?

In our implementation, the ocean model is the first component to run. But the other way round would also work. Before starting with the coupling iterations, small 'pre-runs' are conducted for both components to ensure that all necessary files are available to start the coupling.

> Figure 4: It might be helpful to align the time axes. It's difficult to tell which simulation segments are supposed to coincide with which.

We agree that the shifted time axes are confusing, which has also been pointed out by the second reviewer. We have modified the figure to show that the time axes are aligned (Figure 3 in this document).

> Line 147: Why not extrapolate the values from the centres to the cell boundaries where needed, to minimise the regions of missing data? Would this change anything, or is it implicit in your next point about filling the missing regions?

The algorithm covers this in two steps: first regridding between the grids and secondly filling missing values appropriately. The reviewer is right with the interpretation, that extrapolation to cell boundaries and beyond is part of the second step. Splitting this into different steps, gives the possibility to also take basin boundaries into account. This avoids that values from one basin are extrapolated into a different basin (e.g. across the

[Figure]

Figure 3: *Offline coupling procedure for the PISM-MOM5 setup: the components are run sequentially for the same coupling time step and after each run, variables are exchanged. Temperature and salinity variables from the ocean component MOM5 are used as input fields for the ice component PISM-PICO. Mass and energy fluxes from PISM-PICO output are uniformly applied over the next coupling time step as input to MOM5.*

narrow Antarctic Peninsula where the western basins are much warmer than the eastern basins). To make the difference between regridding and extrapolation more explicit, the manuscript has been updated:

First, the three dimensional output fields (temperature, salinity) are remapped bilinearly from the spherical ocean grid to the Cartesian ice grid. Bilinear regridding is chosen to allow for a smooth distribution of the coarse ocean cell quantities on the finer ice grid.  *Only regions with valid ocean data are filled on the ice grid, which is up to the cell center of the southernmost ocean cell. Areas with missing data need to be filled accordingly (compare grey areas in Figure 5a for example), which is done in the next step. Another option - linear extrapolation into areas with no ocean data coverage by the bilinear regridding scheme - is not applied here as it can lead to unrealistic results.*

> Line 148: It is unclear whether the missing values are filled as a single block of adjacent missing values, or cell by cell. This would lead to slightly different behaviour in the averaging routine.

We apologise for the ambiguity. All grid cells with missing values in the same basin and vertical level are filled synchronously with the same averaged value of all adjoining valid cells. The manuscript has been changed to describe this more precisely:

Secondly, missing values are filled with appropriate data, namely the average over all existing values that are adjacent to grid cells with missing values. This procedure is conducted for each basin and vertical layer*, using the same mean value of adjoining valid cells for all missing grid cells in that region.* Now, the continental shelf area between the ice shelf front and the continental shelf break (see Figure 5a), which is used

*by PICO to calculate the basin mean values of oceanic boundary conditions, is entirely filled with appropriate values.*

> Line 154: What happens to the vertical interpolation if the ocean bathymetry is shallower than the continental shelf depth as seen by PISM?

In this case, the vertical interpolation is omitted and the algorithm chooses the lowermost ocean layer available instead. This has been added to the current version of the manuscript:

*Lastly, the three dimensional variables are reduced to two dimensional PICO input fields which represent the ocean conditions at the depth of the continental shelf. This is done by vertical linear interpolation: for every horizontal grid point, the data is interpolated to PISM's mean continental shelf depth of the corresponding basin. In case the ocean bathymetry is shallower than the continental shelf depth as seen by PISM, the lowermost ocean layer is chosen. An example of the processed input data for PICO is shown in Fig. 5b.*

> Line 163: If I understand correctly, every coastal ocean cell in a given basin receives the same mass of freshwater from PISM (and similarly for energy). Why is this not scaled by area, so that larger ocean cells receive more freshwater? This would be equivalent to distributing the total mass flux evenly over the coastline in the given region, regardless of the details of the ocean grid.

The reviewer is correct that in the original version of the coupling framework as described in the preprint each ocean cell in the same PISM basin receives the same mass of freshwater and energy. This appeared to be a reasonable choice, as usually most of the ocean cells mapped to a given PICO basin have the same latitudes and thus the same area. However, also different ocean cell areas can occur in the same basins, introducing subtle inconsistencies in the distribution of the mapping algorithm. To avoid this, we have adapted the mapping algorithm to calculate the fraction of attribution for each ocean cell weighted by the cell area. Even though the difference is minor in the presented setup, the mapping is more generic now and robust also for different configurations. We thank the reviewer for pointing out this issue.

> Figure 7: It's confusing to compare the different units in (a) and (b). Over what time interval are the fluxes in (a) integrated?

The interval over which the fluxes in Figure 7a have been integrated is a 10 year coupling time step. We apologise that this was not clear and thank the reviewer for addressing this. To enable a better comparison of fluxes between (a) and (b), the labels have been standardised and show the fluxes now both in units $\text{kg/m}^2\text{/s}$. The improved figure including the modified description is shown in this document as Figure 4 (Figure 7 in the preprint).

> Line 171: What happens if the energy flux causes supercooling of the ocean waters? Is this supercooling automatically removed by the sea ice model?

This is correct. The energy flux from ice sheet to ocean component is typically negative, which results in a negative enthalpy flux applied to the ocean component. When this causes a drop of temperature below the local freezing point, frazil ice is formed by the sea ice model.

[Figure]

Figure 4: *Visualisation of (a) PISM mass flux distribution to (b) the MOM5 ocean grid. PISM output variables describing surface runoff, basal melting and calving are aggregated over space and time (coupling time step) to calculate mass and energy fluxes which are processed as input to the MOM5 ocean component as described in Section 4.2. Panel (b) shows the corresponding mass flux distribution on the MOM5 grid.*

> Line 175: Doesn't ignoring these heat fluxes mean the coupled model does not conserve energy? I understand the argument later that the coupler itself conserves energy, during the regridding process, but it appears false to claim that the entire system is conservative (as is implied in line 317).

Indeed the statement that 'conservation of mass and energy is obtained in the coupled system' (l.317) is misleading when referring to the coupling interface only. This work does not and cannot aim at improving the performance or the conservation in the coupled model components. It aims (only) at not introducing additional instabilities or conservation errors in the coupling process. We thank the reviewer for spotting this. The manuscript has been modified accordingly:

*We can assure that conservation of mass and energy is obtained in the  coupler between the ocean and land ice components while having a computationally efficient and flexible coupling setup.*

> Line 188: Explain why the coupling time step is an important parameter. What do you expect would happen if it were too long or too short?

We agree that it should be made clearer why the coupling time step is important and thank the reviewer for pointing this out. A too short coupling time step certainly results in a waste of compute time and disk space for restart and coupling overhead. Too long coupling time steps on the other hand could possibly yield instabilities and might not capture the physical interactions between the components adequately. So far, the manuscript has been modified as follows:

*For the modelling of ice-ocean interactions, the coupling time step is an important parameter that requires careful adjustment, while keeping the different time scales of ice and ocean processes in mind.*  *Too short time steps certainly yield a waste of compute time and disk space for restart and coupling overhead. Too long time steps could possibly yield instabilities and lead to a less accurate representation of ice-ocean interaction processes. Here, only the influence of the coupling frequency on the overall*  *runtime performance is assessed, leaving the examination of physical implications to future work. Two experiments with time steps of 1 and 10 years are compared.*  *, with a total number of 200 and 20 coupling iterations, respectively*  *. The individual coupled component simulations start from quasi-equilibrium conditions.*

As pointed out in our response to the reviewer's comment on model validation above, we will include coupled model results in the revised manuscript that prove the framework's ability to represent the present-day climate in a stable manner over millennial time scales. This will also include a comparison of different coupling time steps and their influence on the physical behaviour of the ice and ocean components.

> Line 200: Clarify that MOM5 runtimes are slightly longer due to a greater fraction of time spent initialising, as with PISM (I assume this is the case).

Yes, the subtle increase in MOM5 execution times is due to the model initialisation, which is done 10 times as often in the yearly coupling compared to the decennial coupling setup. This has been clarified in the manuscript:

*In the experiment using a yearly coupling time step,*  *the elapsed time for all MOM5*  *increase slightly (13 280 s) compared to 10 yearly coupling (12 330 s). The*  *increase is due to component initialisation overhead which occurs 10 times as often as in the decennial coupling configuration. The ocean component postprocessing (9%) and*  *intercomponent processing routines (6%) are taking a bigger share of the total runtime as*  *the number of executions has similarly increased by the factor 10.*

> Line 222: Does this statement mean that MOM5 does not conserve mass? That is concerning, and should be explained further and referenced.

The purpose of this work is to show that there are no mass conservation errors introduced in the coupling process. As we show here, indeed, errors in the coupled model are minimal. However, there seems to be a spurious mass drift in the specific coarse-grid MOM5/SIS *standalone* setup used in our experiments. While we agree with the reviewer that this needs to be investigated further, it would be beyond the scope of this study.

> Line 250: As the Galton-Fenzi reference only refers to an EGU presentation, a better reference for online coupling would be Jordan et al. 2018 (doi:10.1002/2017JC013251) which is already published in JGR.

We thank the reviewer for this suggestion. This is indeed a better reference than the EGU presentation. The reference has been changed in the manuscript.

> Line 257: I'm not sure it's essential for online coupling to have the same timestep for ice and ocean, as ocean models often subcycle different timesteps for different processes (eg barotropic vs baroclinic modes).

Indeed, in a coupled model there are generally several processes running on different time steps inside the different model components. One example, as pointed out by the reviewer, are the barotropic/baroclinic time steps inside MOM5. It would be possible to introduce an additional outer time stepping cycle which wraps the ocean and sea-ice time steps to synchronise to with an online coupled land ice component. But that would require subsequent modifications in the code structure of coupler, ocean, and land ice model, which we can avoid with the offline coupling approach. Furthermore, PISM's adaptive time stepping scheme with all its advantages could hardly be used in that approach.

> Line 273: The ocean mixed layer will rarely extend as deep as the ice shelf front. Ice shelf meltwater entering the ocean at depth would therefore destabilise the water column and deepen the mixed layer, whereas applying this flux at the surface would have the opposite effect. I appreciate that applying meltwater at depth is not trivial in MOM5, but more attention should be given to the possible negative impacts of this design choice on the ocean simulation.

We thank the reviewer for raising this point and agree that the consequences of meltwater input depth needs more consideration. According to Pauling et al. (2016), the spread of freshwater input depth ranges from (close to) surface up to 500m in depth (derived from RTopo-1 dataset, Timmermann et al. 2010). In their simulations a seasonal dependence of whether the input depth is within in the mixed layer or below (especially in summer and autumn) is observed (Pauling et al., 2016, Fig.4). To consider the shortcomings of surface meltwater input in our study in more detail, we have modified the manuscript as follows:

*As described in Section 4.2, the mass and energy fluxes computed from PISM output are given as input to the ocean surface rather than being distributed throughout the water column - a limitation of  MOM5's simplified treatment of all land-derived mass fluxes, including those from ice sheets. This simplification may affect vertical heat distribution and local sea ice formation (Bronselaer et al., 2018) as near-surface input generally makes the vertical column more stratified, whereas input below the mixed layer destabilises the water column, thus enhances vertical mixing and extends the mixed-layer depth (Pauling et al., 2016). A more realistic input depth into the ocean would be the lower edge of the ice shelf front (see start of upper green arrow in Fig. 3; Garabato et al., 2017) which could be determined as the average ice-shelf depth of the last PICO box. *

**Comments by Reviewer 2 (Rupert Gladstone)**

> The paper describes an implemented approach to interactive coupling between an ice sheet model and an ocean model through a reduced complexity ocean cavity model. The project provides a compromise between complexity and efficiency, allowing large scale coupled simulations in which at least some aspects of cavity circulation are represented. This is a useful contribution to global modelling efforts. For the most part, the work is very clearly described and presented. The figures are well prepared and appropriate. I would recommend publication of this paper with some fairly minor modifications.

We thank Rupert Gladstone for taking the time to review our manuscript. The comments are very helpful and we will answer them point by point below.

**General Comments**

> I'm not entirely sure what other models might be competing with the current study in terms of global coupled models of intermediate complexity, but I think maybe the UKESM falls into this category? Are there publications about UKESM and how does it compare to your approach? Perhaps it is significantly more computationally expensive than your setup? I think it uses BISICLES and NEMO for ice and ocean models.

Thanks for raising the point of comparable existing work. This had not been sufficiently addressed in the preprint-version of the manuscript.
There is ongoing effort on coupling the ice sheet model BISICLES to the ocean model NEMO as part of the UKESM development, referred to as UKESM-IS. Unfortunately, we could not find any publication describing more details about the coupled model in general, and especially how the coupling is implemented. A recent study by Gierz et al. (in review, 2020) describes the coupling of AWI-ESM with PISM via SCOPE which is using a comparable approach to our offline coupling framework, although focusing on ice-atmosphere coupling on the Greenland Ice Sheet. As the manuscript would benefit from a discussion of competing and existing ice-climate coupling approaches as also requested by reviewer 1, we will include this in a revised introduction in the resubmitted manuscript.

> The first reviewer recommended including model validation in this paper. Robust model validation is a very large challenge and is not typically included in model description papers. I would prefer to see separate studies present some level of validation rather than try to include it here. Note that this is a highly tunable model, and so a good match to observations over a short time period should be straight forward to achieve. The real challenge will be in quantifying the uncertainty in the model as conditions evolve significantly over long periods of time, and this challenge is well beyond the scope of the current paper. The paper does present some level of model verification, especially regarding conservation, and this is useful.

We agree with the reviewer that fully validating the model would require assessing uncertainties in simulations over long time periods. As we have in the meantime worked towards such a full validation in another study, we have first, promising runs. We will include a short discussion on the runs in the revised manuscript, as the simulations show the spin-up procedure of the coupled setup and that the coupled setup is able to simulate a stable ice sheet and ocean under present-day climate. More detailed analysis and model validation will then follow in a subsequent publication.

> I am aware that some aspects of my review read like an advertisement for the Earth System Modelling Framework (ESMF)! I shoud clarify I am fully independent of ESMF; I'm just an end user. In my experience ESMF is a very robust, user-friendly, well-documented code. So I guess I'm just a fan! It is a valid point though. Existing coupling frameworks (and I talk about ESMF rather than OASIS or others simply because I have the most experience with it) do provide more features and flexibility than the authors convey knowledge of in some places (see also my specific points below).

Thanks for pointing us to these frameworks. See also the replies to the comments below.

> It is very interesting to learn that the PISM intialisation time becomes a significant factor as coupling timestep is shortened. It makes the question "what coupling timestep do you need?" rather important. I couldn't find a comparison of important result metrics between the 1 year and 10 year coupling timesteps. Information about computational cost is given, but is the actual behaviour different? For example, do they give the same total melt rate over the 200 years? Does the ice shelf thickness evolution look the same in both simulations? And grounding line and ice front evolution? You can't really address whether your framework meets requirement 2 until you've established whether or not a 10 year coupling timestep is sufficient.

This is a very valid point and in line with the first reviewer's comment on model validation and coupling time step (line 188). Thanks for addressing this. As mentioned in the response to the comments of the first reviewer, we will include coupled model results in the revised manuscript that prove the framework's ability to represent the present-day climate in a stable manner over millennial time scales. This will also include a comparison of different coupling time steps and their influence on the physical behaviour of the ice and ocean components. With these tests we can re-evaluate whether a long time step of 10 years is sufficient to produce reasonable results and subsequently whether requirement 2 (computationally efficient coupling) is fulfilled.

> I didn't find information about parallelism of the coupling. I infer from this lack of information that the coupling (bash script and file manipulation) all occurs on one processor. If this is the case, can you comment on how many processors you intend to use for production runs, and whether this might become a bottle neck for larger parallel simulations? For example, the ocean postprocessing and intermodel processing took 15

Currently the data processing for the coupling is implemented sequentially. Compared to the compute times of component models it is so small that it appears not worth the effort to parallelise it. For production runs we are currently using 32 cores. For higher grid resolutions, the interpolation processing similar to the integration time of the components will require longer run times. We do not expect the interpolation to become a bottleneck in this case (in particular since we rely on bilinear regridding and not conservative regridding which is much more expensive). If it becomes a bottleneck, we agree that this would be a good way to improve performance.

> The Earth System Modelling Framework (ESMF) community adopts some terminology that I find quite beneficial in that it offers clarity in certain areas that can otherwise become slightly confusing. Individual models in a coupled system are referred to as components. This avoids confusion between the coupled model and the individual ice and ocean models. I like this clarity and I think it would be nice the the authors adopt it, but I do not wish to make this a requirement, just a suggestion.

We agree that addressing the coupled setup as 'model' and the individual ice and ocean submodels as 'components' introduces more clarity. We thank the reviewer for this suggestion and will update the manuscript accordingly.

**Specific Comments**

> Line 107: I don't quite understand the use of two horizontal dimensions. If I understood right, PICO just represents the overturning circulation. How do two horizontal dimensions come into play here? Or perhaps I should go and read Ronja's 2018 paper on PICO (lazy reviewer!)!

The implementation of PICO includes a routine to process the 2-dimensional input fields. It averages the input over the continental shelf in front of the ice shelf cavity in each basin. This makes it easier to have changes in the ice front position feed back onto the ocean forcing during the transient simulation. The input for the box model solutions is then a single value in each ice shelf. To make this a bit clearer, we changed "required" to "uses" in the manuscript:

*PICO  uses two dimensional (horizontal) input fields, namely temperature and salinity of water masses that access the ice-shelf cavities, to calculate melting and refreezing at the ice-ocean interface, as illustrated in Fig. 2.*

> Lines 113-114: This is also known as "sequential coupling".

Thanks for this remark, it has been added to the manuscript:

*This is one motivation among others to use an offline coupling (also known as sequential coupling) approach to exchange the fields between the two components.*

> Line 115: In this context, does an "integration step" mean running the model for a coupling time step? Can you clarify this in the text?

Exactly. The term "integration step" has been used as a synonym for "coupling time step" in this context. To avoid confusion, this has been changed and clarified in the manuscript:

* In the offline coupling procedure, one component is first run for the period of a coupling time step. Then the output is processed and provided as input or boundary condition to the other component, respectively. Using the modified input, the models are restarted from their previous computed state.*

> Line 118: "the last of these averaged fields" is a slightly confusing expression. Presumably PISM receives the average of the fields over the full (10 years in this example) coupling timestep? But this isn't clear to me from the chosen wording.

In the initial implementation of the framework as described in the preprint, PISM received the last record of the annual mean diagnostic output for ocean temperature and salinity. We have changed the implementation to

pass ocean temperature and salinity to PISM that are averaged over the full coupling time step of the last ocean model execution. We thank the reviewer for raising this point. The changes in the manuscript are the following:

*For example, MOM5 runs for 10 years and writes annual mean diagnostics fields of temperature and salinity. PISM receives the  temporal average of these fields over the coupling time step as boundary conditions for PICO, and is then integrated for the same 10 year period. Melt water and energy fluxes derived from PISM output are aggregated over the coupling time step. The resulting fluxes are then added as external fluxes to the ocean over the course of the next integration period. To avoid shocks in the forcing, they are distributed uniformly over the entire coupling time step.*

> Figure 4: There appears to be a temporal offset between $t_{ice}$ and $t_{ocean}$ in the way that the Figure is presented. But the text suggests that the time period over which the two components are integrated should not be offset (line 113 refers to the "same model time"). Can you clarify this? It is not clear to me what is meant be "Sharing the same time axis", perhaps this is related to my question?

In our implementation, the model components share the same time axis. We admit that this can be understood differently through the way it is presented in Figure 4 of the preprint. The same point was also raised by the first reviewer, and we have modified the figure to make clear that the time axes are aligned (Figure 3 in this document).

> Line 134: I don't know what "entanglement" means in this context.

We acknowledge that the chosen wording was not clear and thank the reviewer for pointing this out. By "horizontal grid entanglement" we mean the way that both horizontal grids are intertwining such as there exists overlaps as well as spatial gaps of both model grid domains. The corresponding text has been rephrased in the manuscript:

*The ice and ocean  components operate on independent, non-complementary computational grids. The inset of Fig. 3 shows that there are both, spatial gaps and overlaps, between the ocean grid cells and the ice extent represented by PISM. As the ocean grid is much coarser than the ice grid and MOM5 cells are either defined entirely as land or ocean (no mixed cells allowed), inconsistencies in the  exchange of quantities between the two grids are unavoidable, requiring careful consideration of  data regridding.*

> Line 139: Is "surrounding ice sheets" a typo in this context? I mean, the Zwally basins divide up the Antarctic Ice Sheet... perhaps you mean shelves not sheets here?

This is a typo in the text. We thank the reviewer for spotting this. The text has been corrected accordingly:

*They are based on Antarctic drainage basins defined in Zwally et al. (2012) and extended to surrounding  ice shelves and the Southern Ocean, see Fig. 5b.*

> Lines 148-149: If you use only adjacent cells to populate missing values, presumably you iterate until all cells have a value? I mean there must, to start with, be plenty of cells that are not adjacent to a cell with a value.

We apologise for the ambiguity. All grid cells with missing values in the same basin and vertical level are filled synchronously with the same averaged value of all adjoining valid cells. The manuscript was modified to describe this more precisely:

*Secondly, missing values are filled with appropriate data, namely the average over all existing values that are adjacent to grid cells with missing values. This procedure is conducted for each basin and vertical layer, using the same mean value of adjoining valid cells for all missing grid cells in that basin. Now, the continental shelf area between the ice shelf front and the continental shelf break (see Figure 5a), which is used by PICO to calculate the basin mean values of oceanic boundary conditions, is entirely filled with appropriate values.*

> Lines 159-160: I'm fairly confident that ESMF's "common" regridding algorithms include masked nearest neighbour remapping options that are very simliar to what is described here. I intend to use these for remapping subglacial outflow from an ice model to an ocean model, though I haven't actually implemented this yet, and it looks like the required functionality to do this in a mass conserving way is already in place.

Thanks for pointing out that the ESMF regridding tools support our use case. By using the PICO basins, we have the advantage that the ocean cells and the PICO forcing match nicely in terms of fluxes distributed in either way. We have made the following changes to the manuscript:

* There are large areas of the PISM domain that are not overlapping with valid MOM5 ocean cells (see white areas in Fig. 1b and inset in Fig. 3). To ensure mass and energy conservation, we introduce a new mechanism for the coupled system which maps every southernmost ocean cell of the MOM5 grid to exactly one PICO basin (see Fig. 6).*

> Section 5.1. I can't quite make the numbers add up here. The run with a 1-year coupling timestep takes 22700s. The ocean post-processing, interprocessing and PISM percentages given add up to 35

We understand that confusion is caused by not mentioning all percentages in the text and thank the reviewer for bringing this up. For clarification, we added a table of measured runtimes and their corresponding total percentages in the appendix (shown here in Table 1).
Currently the individual percentages in the table do not add up to the total runtime, as book keeping for the coupling post processing (concatenating the individual output files) is missing in the table. This will be corrected in the revised version of the manuscript.

> Line 193: I presume "total runtime" is the elapsed time not the cpu hours? i.e. the total computational time would be this number times 32?

The reviewer is right with his interpretation. With "total runtime" we are referring to the wall-clock time or elapsed time. CPU hours, if defined as the sum of elapsed time on every allocated CPU cores, would be the

Table 1: Runtimes of coupled PISM-MOM5 setup for 200 years model time using 32 cores. PISM runtimes include PICO and MOM5 runtimes include SIS and FMS components.

| | 1 year coupling | | 10 year coupling | |
| routine | time [s] | ratio [%] | time [s] | ratio [%] |
| --- | --- | --- | --- | --- |
| total | 22652.15 | 100.00 | 13700.67 | 100.00 |
| preruns | 24.46 | 0.11 | 32.57 | 0.24 |
| preprocessing | 40.55 | 0.18 | 41.10 | 0.30 |
| MOM runs | 13282.75 | 58.64 | 12330.02 | 90.00 |
| MOM postprocessing | 2057.29 | 9.08 | 252.25 | 1.84 |
| PISM runs | 4386.67 | 19.37 | 661.63 | 4.83 |
| MOM-to-PISM processing | 1214.94 | 5.36 | 183.78 | 1.34 |
| PISM-to-MOM processing | 90.74 | 0.40 | 10.90 | 0.08 |

wall-clock time times 32 in this case. The manuscript has been changed to describe more precisely what we mean with 'total runtime':

*The elapsed total runtime (wall-clock time) required for 200 years model time is 22 700 s and 13 700 s with a coupling time step of 1 and 10 years, respectively.*

Equation 1: Surely mass has dimensions of mass and smb has dimensions of mass/time. So how can you simply add these? I don't understand why there would be rates in this equation. Surely the total mass is the sum of the mass from each component at any given moment in time? The same comment for lines 214-215. How can you subtract a flux from a mass? I can't make sense of it!

Line 222: It is not clear from this description whether the dimensions of dosi should be mass or mass/time.

We admit that the chosen expressions are miss-leading and apologize for the confusion. We renamed $smb_{\mathrm{osi}}$ and $smb_{\mathrm{li}}$ to $m^{\mathrm{s}}_{\mathrm{osi}}$ and $m^{\mathrm{s}}_{\mathrm{osi}}$ respectively as they are not describing the actual mass balance (mass/time) but the cumulative, integrated surface mass balance flux over space and time of the ocean-sea ice component and the land ice component, which is a mass. The same applies to the model mass drift, which was renamed from $d^{\mathrm{d}}_{\mathrm{osi}}$ to $m^{\mathrm{d}}_{\mathrm{osi}}$ as this is also a mass. The manuscript was changed accordingly:

*To check that the total amount of mass and energy stocks is constant in the coupled system over the model integration, we assess virtual quantities. Those are obtained by subtracting the masses applied through surface fluxes from the total mass and energy stocks calculated in the model (see Eq. (1) for mass). If the virtual model mass across the model  components $m_v$ is constant with fluctuations in the order of machine precision, as denoted in Eq. (2), conservation of mass is achieved.*

$$m_{\mathrm{v}} = (m_{\mathrm{o}} + m_{\mathrm{si}} - \underline{m}^{\mathrm{s}}_{\mathrm{osi}} - \underline{m}^{\mathrm{d}}_{\mathrm{osi}}) + (m_{\mathrm{li}} - \underline{m}^{\mathrm{s}}_{\mathrm{li}}) \tag{1}$$

$$\frac{d}{dt} m_{\mathrm{v}} \sim 0\, Gt/a \tag{2}$$

*The masses of the ocean, sea ice and land ice  components are represented by $m_o, m_{si}$ and $m_{li}$ respectively, while  $m^s_{osi}$ and $m^s_{li}$ denote the cumulative, spatially integrated surface mass balance flux of the ocean-sea ice  component MOM5/SIS and the land ice  component PISM, respectively. The internal model drift of mass in the coarse-grid MOM5/SIS setup is described by  $m^d_{osi}$ ($\approx 4 \cdot 10^{15}\, kg$  accumulated over 200 years) and needs to be considered in the computation of virtual model mass in Eq. (1).  All terms in Eq. (1) are quantities of mass with the temporal*

*resolution of the coupling time step. The relative mass conservation  error $e_{rel}^m$ is calculated as fluctuations of the virtual model mass compared to its temporal mean $\overline{m_v}$, noted in Eq. (5) .*

$$e_{\mathrm{abs}}^{\mathrm{m}} = m_{\mathrm{v}} - \overline{m_{\mathrm{v}}} \tag{3}$$

$$e_{\mathrm{rel}}^{\mathrm{m}} = e_{\mathrm{abs}}^{\mathrm{m}}/\overline{m_{\mathrm{v}}} \tag{4}$$

$$e_{\mathrm{rel}}^{\mathrm{m}} = \frac{m_{\mathrm{v}} - \overline{m_{\mathrm{v}}}}{\overline{m_{\mathrm{v}}}} \tag{5}$$

> Figure 9: To put this apparently small error into context, it would be useful to give some indication of how much mass is transferred between ice and ocean. I think the total mass transfer is probably a more relevant figure here than the total ice or ocean mass. We need to know that the error measures are small not only compared to the total mass of the coupled system, but also small compared to the amount of mass being transferred between ice and ocean over the integration period.

We thank the reviewer for this very good suggestion. A subgraph has been added to Figure 9 (Fig. 5 in this document) which compares the fluctuations of the virtual coupled mass to the mass transferred between the components. The text and the figure's caption has been changed accordingly:

*The relative mass conservation error $e_{rel}^m$  is shown in Fig. 5a for 200 model years with a yearly coupling time step. Regarding the order of magnitude of land ice mass $\mathcal{O}(m_{li}) = 10^{19}\,kg$ which is given in single precision ($\approx$ 7 decimal digits) output format and the order of magnitude of ocean and sea ice mass $\mathcal{O}(m_o + m_{si}) = 10^{21}\,kg$, given in double precision ($\approx$ 16 decimal digits) format, the shown fluctuations in the order of $10^{-9}$ are reasonable. As the relative mass error does not show a trend, no systematic error is introduced through the coupling procedure. In Fig. 5b the fluctuations of virtual model mass is also compared to the mass flux between the land ice and ocean component ($m_x$), which is in the order of $\mathcal{O}(10^{-3})$.*

> Line 239: I don't think you can make this unqualified statement that the framework fulfills all three requirements. 1 and 3, yes, sure, but requirement 2 is really only partially fulfilled, as you continue on to discuss in the following paragraph.

We thank the reviewer for the comment and agree: the question of a computationally efficient coupling framework (fulfillment of requirement 2) is best reassessed with the evaluation of the coupled test runs that we will conduct as described above. The resubmitted manuscript will address this accordingly.

[Figure]

Figure 5: *(a) Relative error of virtual mass progression in the coupled ice-ocean system which excludes mass changes applied through surface fluxes and the MOM5/SIS internal model* component *drift. (b) A comparison of virtual mass fluctuations to the mass exchanged between ocean and land ice components ($m_x$). (c) Relative error through remapping energy flux from PISM to MOM5 grid, where $e_i$ and $e_o$ describe the transferred energy fields (unit W) on the land ice and ocean grid respectively. $\Sigma e$ is the spatially aggregated energy over the whole grid domain.*

Paragraph 248-267. This paragraph completely omits to discuss alternative forms of online coupling in which a single executable links to component runtime libraries and fulfills a role equivalent to your bash script as a master (or parent) program. The implication in this paragraph is that one must choose between an offline coupling in which components are called independently and an online coupling in which one component must be the master and the other the slave. But this is not the choice that a coupled model developer faces, there are many more options available. For example, the Earth System Modelling Framework (ESMF) offers full flexibility in this sense. The developer can choose whether to create a new parent routine or to establish one component as master. You might want to cite Gladstone et al GMDD FISOC paper (published in June 2020 as a discussions paper and now accepted pending minor revisions for GMD), which is essentially an online equivalent of your coupling structure, with a new (Fortran in this case) master program that calls the child components (which have been made ESMF compatible and compiled as libraries). This approach also allows flexibility in terms of switching between components (indeed we currently have a choice of two ocean models coupled through FISOC, and two furtehr ice sheet models are in the process of being incorporated). Of course there is still some overhead in terms of ensuring each component is compatible with the coupling framework (ESMF in the case of FISOC), so it is a longer development path than your bash script plus file manipulations, but perhaps not as onerous or restrictive as implied in the current version of the text.

We thank the reviewer for the remark. We appreciate the possibilities of ESMF and FISOC architectures and are happy to learn more about them in the future. The variant proposed by the reviewer (new main program which as acts as coupling master and links to component runtime libraries) has mostly the same disadvantages as using one of the components as master and the other as slave, linked together into one online coupling executable. In both cases it is necessary to modify at least one of the component model codes in a suitable way so that it can be linked into the executable program. There might be model codes for which such modifications were already done, or anticipated, by others before, and thus are easier to link into a common executable. In fact, MOM is prepared to be linked into coupler frameworks like FMS or Access, but PISM is not. In our work, we preferred to avoid writing a new main program and maintaining interfaces altogether, by using the described approach via a master shell script, with all the described advantages.

To include the reviewer's reasonable argument about different possibilities for online coupling architecture (master-slave structure), we have made the following changes to the manuscript:

*The chosen offline coupling framework executes the two different components alternately and independently and takes care about redistributing the input and output files across the components as explained in Section 3. However, it is also conceivable to adopt an online coupling approach (also called synchronous coupling), where the ice and ocean component code are consolidated into one code structure. The exchange of variables  between both components can subsequently take place through access to the same shared memory instead of writing the required variables to disk and reading from there again, as it is done in offline coupling.  This approach is for instance used by Jordan et al. (2018). A comprehensive framework for online coupling of ocean and ice components is described in Gladstone et al. (2021). This coupling approach is especially powerful for high resolution, cavity resolving ice-ocean coupling, where frequent updates of the ice-shelf cavity geometries and corresponding melt rates are important. However, a prerequisite for online coupling is the adaptation of the  standalone models for interactive execution of subroutines through a defined (external) interface. In the given case of coupling PISM and MOM5, this means that at least one of the two programs' code structure needs major modifications and modularisation to equip the individual component parts like initialisation, time stepping routine, disk I/O, stock checking, etc with suitable interfaces. This is independent of the chosen online coupling design (incorporating one code structure into the other or creating a new master program which governs both components). Synchronisation of the PISM adaptive time step and the fixed ocean component time step would be a further issue, also keeping in mind that the comparably small ocean time step of a few hours is not applicable for the ice component: PISM can have a time step of around 0.5 years close to equilibrium with 16 km resolution due to the longer characteristic timescales of ice dynamics.*

> Line 256. ESMF (for example) handles all parallel regridding in an efficient manner. So long as component mesh and field information can be made available in ESMF runtime data structures (which I acknowledge requires some coding and may not be trivial), the regridding between different partitioning is all handled automatically.

Thanks for the remark. The referenced sentence has been removed from the manuscript (see comment above).

> Line 260. The C/Fortran issue is a fairly small technical issue. There are plenty of codes around that use both at runtime.

We agree, and mention it just for completeness.

> Lines 269 - 274. The physical implications of this issue could be fairly interesting. Could the input ever come in beneath the turbulent mixed layer? Could the input of fresh water at the surface have a stabilising effect that would not occur if it was mixing up from lower down? Do you have any plans to investigate this further? This could all raise interesting questions that have been brushed over very lightly here.

This has also been noted by the first reviewer and we appreciate both comments on this issue. According to Pauling et al. (2016), the spread of freshwater input depth ranges from (close to) surface up to 500m in depth (derived from RTopo-1 dataset, Timmermann et al. 2010). In their simulations a seasonal dependence of whether the input depth is within in the mixed layer or below (especially in summer and autumn) is observed (Pauling et al., 2016, Fig.4). To consider the shortcomings of surface meltwater input in our study in more detail, we have modified the manuscript as follows:

*As described in Section 4.2, the mass and energy fluxes computed from PISM output are given as input to the ocean surface rather than being distributed throughout the water column - a limitation of  MOM5's simplified treatment of all land-derived mass fluxes, including those from ice sheets. This simplification may affect vertical heat distribution and local sea ice formation (Bronselaer et al., 2018) as near-surface input generally makes the vertical column more stratified, whereas input below the mixed layer destabilises the water column, thus enhances vertical mixing and extends the mixed-layer depth (Pauling et al., 2016). A more realistic input depth into the ocean would be the lower edge of the ice shelf front (see start of upper green arrow in Fig. 3; Garabato et al., 2017) which could be determined as the average ice-shelf depth of the last PICO box. *

> Lines 285 - 288. This is actually a very important issue, especially because you intend this framework to be applied to long timescale simulations. You mention that work is in progress, but can you say a few lines about how this will be implemented? Can you also ive an example or two of the how you intend to use the model in its current form so that the reader can start to envisage how much of an issue (or not) the lack of evolving active ocean domain is?

Thanks for this comment. We have adapted and extended the paragraph to address the request for more details on the described mechanism (see below). For the multi-millennial coupled runs that prove the stability of a coupled configuration for present-day conditions (to be included in the resubmitted manuscript), the described mechanism is not required due to a lack of significant mass changes between ocean and ice. Future analysis of long term interactions will include a more detailed consideration of this mechanism.

Changes in the manuscript:
~~During long simulations where glacial and interglacial periods are alternating, large amounts of water are transferred between oceans and ice sheets. Through significant changes in the sea level, whole ocean cells can be subject to wetting or drying. The land-ocean mask needs to be adapted accordingly during the simulation including a meaningful way to handle mass and energy stocks. The current framework is not capable of managing such changes yet, but development is currently in progress.~~ *The waxing and waning of ice sheets*

*on glacial-interglacial time scales causes transfer of large amounts of water between the oceans and land ice sheets. Significant changes in sea level (120-135 meters below present during the last glacial maximum (Clark and Mix, 2002)) have large impacts on coast line positions. The response of the solid Earth component to changes of ice-sheet mass has a similar effect. During long simulations the land-ocean mask needs to be adapted accordingly. As MOM5 cannot handle mixed ocean-land cells, which would allow for a smooth adaption of a changing coast line, major changes in the land-ocean mask need to be performed during a transient simulation. This requires careful considerations like the initialisation of newly flooded cells and implications concerning mass and energy conservation as well as model stability. The development of a sea-level based dynamic ocean domain adaptation which applies the described changes to new ocean restart conditions is currently under way and will be incorporated in the described coupled setup in the future.*

> Lines 293-295. These lines seem to imply that tuning PICO will somehow make up for the lack of a full representation of the complex 3D ocean circulation over the continental shelf and under ice shelves. This will not, in general, be the case. It is clear that this model is a compromise approach, a model of intermediate complexity with the benefit of efficiency. This has value; you don't need to try too hard to defend or justify limitations of PICO. I would prefer to see the limitations presented directly without implying that they can be overcome (through tuning for example).

We thank the reviewer for this thoughtful comment and have modified the manuscript as follows:

*In the coupling framework, ocean input for PICO is averaged over the entire basin, not taking into account horizontal differences such as cavity in- and outflow regions and possible modifications of water masses on the continental shelf. Furthermore, complex processes determine what water masses make their way from the open ocean onto the continental shelf and to the grounding lines (Nakayama et al., 2018; Wåhlin et al., 2020). However, in our coarse grid setup of MOM5, bathymetry and circulation on the continental shelf are only partly represented (see also Fig. 1b). PICO currently does not represent circulation patterns besides the vertical overturning circulation, and they hence need to be considered in the tuning process of the PICO parameters.The presented coupling framework is characterized by a reduced complexity approach. This is reflected for instance in the basin wide averaging of PICO input which does not account for horizontal differences such as cavity in- and outflow regions or modification of water masses on the continental shelf. Similarly, the complex processes determining whether upwelling Antarctic Circumpolar Deep Water reaches the continental shelf and the grounding lines (Nakayama et al., 2018), can only be partly represented due to the coarse bathymetric features of the MOM5 grid (see also Fig. 1b). However the intermediate complexity of the coupled system enables ocean simulations on a global domain, opening possibilities to study interactions, feedbacks and possible tipping behaviour on millennial time scales.*

> 311. "appropriate" is not well defined in this context, and perhaps not yet fully justified. How about just "intermediate" instead?

We agree that this is a better phrasing and have changed the manuscript accordingly:

[revised manuscript text omitted]

---

## Author Response (AR1)

**Author comment on RC1 and RC2 of gmd-2020-230**

Moritz Kreuzer Ronja Reese Willem Nicholas Huiskamp Stefan Petri Torsten Albrecht Georg Feulner Ricarda Winkelmann

> March 4, 2021 updated: April 16, 2021

This statement is addressing the reviews of the discussion paper "Coupling framework (1.0) for the ice sheet model PISM (1.1.1) and the ocean model MOM5 (5.1.0) via the ice-shelf cavity module PICO" (gmd-2020-230) in Geoscientific Model Development (GMD).

This document is an updated version of the Author comment submitted to GMD on March 4th, 2021. Additional changes are:

- A revised and reorganised introduction (Section 1) which now includes a comprehensive discussion of existing ice-ocean coupling approaches and explains the novelty of the approach presented in our manuscript.
- A more detailed description of the PISM setup in use (Section 2.1).
- Updated Figures 6 and 7 (Fig. 5 and 6 in this document), which now correctly display the tripolar ocean grid in high northern latitudes.
- Section 5 has been renamed from 'Results' to 'Evaluation'.
- An updated 200 year benchmark including concatenation of files at the end of the coupled runs (Section 5.1). Figure 8, Table A1 and text in Section 5.1 have been adapted accordingly.
- Section 5 has been extended by Subsection '5.3 Coupled runs for present-day conditions' which includes results of coupled runs for 4000 years, showing that the coupled system remains stable under present-day conditions.
- Updated Zenodo repositories for the coupling framework, PISM and the used input files.

We thank the editor Steven J. Phipps, the first anonymous reviewer and the second reviewer Rupert Gladstone for their interest and time reviewing our manuscript and their thoughtful, constructive and very helpful comments and feedback. We provide detailed responses to all comments below, where the reviewers remarks are displayed as indented quotes. Changes in the manuscript (italics) are given where appropriate, showing old text in red and new text in blue color.

**Comments by Reviewer 1**

"Coupling framework (1.0) for the ice sheet model PISM (1.1.1) and the ocean model MOM5 (5.1.0) via the ice-shelf cavity module PICO" by Kreuzer et al. describes the software implementation of an ice-sheet/ocean coupler, designed for ocean models that do not resolve ice shelf cavities. This manuscript details the algorithm by which data is exchanged between the two models on a basin-averaged basis, while ensuring conservation and a reasonable computational overhead. The scientific validation of the coupled system is left for another paper, so the realism of the model is not discussed.

We thank the reviewer for taking the time to review our manuscript. The comments are very helpful and we will answer them point by point below.

**General Comments**

Coupling ocean and ice-sheet models via PICO is a worthwhile idea, as it would fill a gap in coupled modelling. If successful, it would enable CMIP-style models to include evolving ice sheets without having to explicitly resolve ice shelf cavities, which would open a lot of doors for long-timescale simulations. The coupling algorithm described here is well explained and logically designed. I have a few questions about the coupler which should be addressed for clarity (see my specific comments below).

However I am concerned at how this paper does not show whether the coupled system can actually produce reasonable results, which are acceptably realistic and stable for the present-day climate. I find it acceptable to separate the software description from the model validation, but only if the model validation is submitted as a companion paper at the same time. Without this assurance, I worry that the authors have not yet tested the realism of the coupled system.

What if there are insurmountable challenges which make the premise of the coupling unusable? For example, Figure 5 suggests that the ocean temperatures being passed to PICO are far too warm, with no continental shelf temperatures below about 0.25C. In reality, we know that large regions of Antarctica have inflow into ice shelf cavities consistently around -1.9C, as a result of sea ice formation. Effectively, coupling with this MOM configuration will turn all ice shelf cavities into warm cavities like the Amundsen Sea, but we know that many/most cavities are cold cavities. Surely this would lead to unacceptably high basal melt rates and grounding line retreat.

What if these challenges can only be overcome by substantially changing the coupling design, at which point this paper becomes out of date? While I applaud the authors for their worthwhile efforts in this model development, I worry it is premature to publish before we know whether this approach will work. It might be best to save this paper up until the scientific validation has been completed and written up. At that point, it will be a very nice submission to GMD.

Thanks for addressing the very important point of model validation. We agree that the coupling framework can only be useful if it is able to reproduce a stable coupled setup for the present-day climate. In the originally submitted manuscript (lines 296-307), we suggested a bias correction of the modelled ocean properties with respect to present-day observations as used for example in the ISMIP6 project (Jourdain et al., 2020). We here follow this approach in coupled simulations. For validation of the framework, we included coupled model results in the revised manuscript that show the framework's ability to represent the present-day climate and ice sheet state in a stable manner over millenial time scales. We want to note that the warm bias in ocean temperatures in our model can probably be attributed to the coarse model resolution, however, such bias has also been found in CMIP5 models with higher resolution (Heuzé et al., 2013; Barthel et al., 2020).

We have added the following text to the manuscript:

**5.3 Coupled runs for present-day conditions**

[revised manuscript text omitted]

My other major comment is regarding the introduction. I feel that a stronger and clearer case could be made for why this coupling advance is needed. I would love to see a revised and reorganised introduction, which more clearly lists the major categories of models which already exist (global coupled models with fixed ice sheets; standalone ice sheet models with highly simplified ocean forcing; high-resolution ocean models with static or dynamic ice shelf cavities) and why they are not suitable for ice-sheet/ocean coupling on millennial timescales. This would make it more clear to the non-expert reader why PICO coupling is a major advance.

We thank the reviewer for this constructive comment and agree that the introduction could be improved with a more comprehensive discussion of existing coupling approaches including the novelty of our approach using PICO. We have included a revised and reorganised version of the introduction in the manuscript:

[revised manuscript text omitted]

**Specific Comments**

Title: Is it necessary to include the version numbers for PISM and MOM5, as well as the coupler? This detracts from readability, and it seems like the coupling should be more or less independent of the specific model versions. Also, the title should specify that the coupling is specific to the Antarctic Ice Sheet.

The title includes the version numbers to meet the requirements of GMD model description papers1. To reflect the Antarctic application of the described coupling, we changed the title of the finalised manuscript to "Coupling framework (1.0) for the ice sheet model PISM (1.1.44) and the ocean model MOM5 (5.1.0) via the ice-shelf cavity module PICO in an Antarctic domain".

Abstract, line 8: "Earth system" should be changed to "ocean", since this manuscript doesn't discuss coupling between the ice sheet and other climate model components.

We agree with the reviewer that the suggested wording is more precise. The text has been changed accordingly.

Line 24: The discussion of ocean forcing in paleoclimate should be expanded, as many of the readers may be unfamiliar with this.

Thanks for making this helpful suggestion. This has been reflected in the revised introduction (see also above):

Ocean forcing also plays an important role in the paleo-climatic context (Hillenbrand et al., 2017; Lowry et al., 2019). Ocean forcing has also been identified as playing a major role in past changes of the Antarctic Ice Sheet. Evidence that the Holocene retreat of the West Antarctic Ice Sheet was driven by warm water intrusions onto the continental shelf was provided by paleo proxy data analysis of Hillenbrand et al. (2017) and supported by ensemble modelling for the Ross Embayment (Lowry et al., 2019).

Line 26: "prescribed ice-sheet configurations" is not clear - does this mean there is a fixed ice surface topography and no simulation of ice dynamics?

The reviewer is correct in their interpretation. We apologise that this was not sufficiently clear. The passage has been clarified in the revised introduction (see also above):

 ${}^{1}https://www.geoscientific-model-development.net/about/manuscript\_types.html\#item1$

The standard set of experiments for the Coupled Model Intercomparison Projects (CMIP) are performed by Atmosphere-Ocean General Circulation Models which use fixed, non-dynamic ice sheet configurations and have thus only a limited representation of the aforementioned interactions and feedbacks (category 1; e.g. Eyring et al., 2016)

Line 30: I suggest changing "circulation in ice-shelf cavities" to "ice pump", as PICO doesn't explicitly simulate cavity circulation either.

Thanks for making this suggestion. We agree that the previous phrasing implied that PICO would explicitly simulate ice-shelf cavity circulation, which it doesn't. After reorganisation of the introduction, the corresponding text now reads (see also above):

These coupled setups are using, if they consider ocean forcing on the ice sheet, simple melt parameterisations that do not take the circulation in ice-shelf cavities into account, which is important to estimate realistic melt rates. In addition to the melting at the ice-ocean interface, the Potsdam Ice-shelf Cavity mOdule (PICO; Reese et al., 2018) minics the large-scale overturning circulation in ice shelf cavities.

Line 38: Note that Scenario A1B is from CMIP3, not CMIP5/6.

Thanks for pointing this out. This text segment is not included anymore in the revised introduction.

Line 40: It is not clear that "ice-sheet/ocean interactions" here refers to fully coupled ice-sheet/ocean models including ice dynamics. There are many ocean models which simulate ice shelf thermodynamics alone, with static cavities, which is an important distinction from the types of models I think you are referring to.

We agree that the chosen wording leaves room for interpretation and thank the reviewer for addressing this. This is in line with the reviewer's major comment about reorganising the introduction with a more detailed discussion of existing ice-ocean coupling models and approaches. The revised introduction now gives a general overview and explanation of the different ice-ocean coupling approaches (category 1-5), see above.

Line 46: Donat-Magnin et al. (2017) is not a coupled ice sheet-ocean model, but rather has static cavities as above, and prescribed a different grounding line for a sensitivity experiment. I would suggest just taking this reference out.

We thank the reviewer for spotting this error. The reference has been removed as suggested.

Line 60: Surely it is only 2-dimensional ice sheet models, with SSA or similar? I equate 3-dimensional with full Stokes, which I don't think PICO has been coupled to.

We agree that SSA models could be called 2D, however, PISM is a hybrid model that employs a superposition of the SSA and SIA. Since the SIA is vertically non-uniform, PISM is a 3D model with a 3-dimensional field of ice velocities. It also solves thermodynamics on a 3-dimensional grid. The restructuring of the introduction, made the cited sentence obsolete (see above).

Line 60: Change "approximates" to "parameterises"

This has been changed as requested.

Line 100: I suspect "poles" and "equator" are the wrong way round - usually meridional resolution is finer at the poles, to follow the convergence of zonal resolution. Coarser meridional resolution at the poles, as the text says, would lead to a strange cell aspect ratio.

The description in the manuscript is correct, see also the model description paper from Galbraith et al. (2011, Appendix A.b). For illustration we have added a figure of the meridional resolution of the MOM5 ocean grid in use, see Figure 2. A different point (which hasn't explicitly mentioned so far) is that the grid is of tripolar structure to avoid a singularity at the north pole. This information has been added to the manuscript:

For this study, MOM5 is used with a global coarse grid setup from Galbraith et al. (2011, see Fig. 1b): the lateral model grid is  $3^{\circ}$  resolution in longitude (120 cells) and in latitude it varies from  $3^{\circ}$  at the poles to 0.6° at the equator (80 cells). It makes use of a tripolar structure to avoid the grid singularity at the North Pole (Murray, 1996).

Line 101: Is p\* the same as z\* (which is a more common vernacular for ocean modellers)?

 $p^*$  and  $z^*$  are not synonymous, but similar coordinate systems. While  $p^*$  is a pressure-based coordinate extending over the time-independent range  $0 \le p^* \le p_{bottom}$ ,  $z^*$  is similarly defined in depth space, spanning  $-H \le z^* \le 0$  where -H equals the local ocean depth. More information about the MOM5 internals can be found in the manual2.

Line 102: What is the vertical resolution of the thickest layers?

As the vertical resolution increases with depth, the lowermost vertical layer is the thickest. It has a maximum resolution of ~ 513 dbars which translates to a thickness of 511 m when using hydrostatic water pressure conversion with density  $\rho = 1023.6 \text{ kg/m}^3$  and  $g = 9.80665 \text{ m/s}^2$ . As described in Galbraith et al. (2011, Appendix A.b), the model employs partial bottom cells to allow a more realistic representation of the bathymetry. The vertical resolution of the MOM5 grid in use is shown in Figure 3. To make this more clear in the manuscript, the text has been extended:

The vertical grid is defined using the re-scaled pressure coordinate  $(p^*)$  with a maximum of 28 vertical layers. The uppermost eight layers are approximately 10m thick, gradually increasing for deeper cells to a maximum of ca. 511m. The vertical resolution in depths relevant for ice shelf cavities is between 50 and 180m. The lowermost cells can have a reduced thickness to account for ocean bathymetry with partial cells.

Line 113: The text describes the models as run in alternating order. Is it possible to run them simultaneously on different processors? This would save on walltime, although the CPU hours wouldn't change.

<sup>2https://mom-ocean.github.io/assets/pdfs/MOM5\_manual.pdf, last accessed: 4th March 2021

Simultaneous runs introduce a lag of one coupling time step in the exchange of variables between the models. While this can be acceptable, a concurrent setup makes most sense, when both models require roughly the same computation time. Otherwise one set of processors will be idle most of the time which is bad from an efficiency perspective. In the 10 year coupling setup, the major share of runtime is consumed by the ocean model (see Section 5.1) and a concurrent setup would hardly reduce the runtime.

Line 114: Figure 4 suggests that MOM is the first model to run. Is this always the case, or can PISM start the chain?

In our implementation, the ocean model is the first component to run. But the other way round would also work. Before starting with the coupling iterations, small 'pre-runs' are conducted for both components to ensure that all necessary files are available to start the coupling.

Figure 4: It might be helpful to align the time axes. It's difficult to tell which simulation segments are supposed to coincide with which.

We agree that the shifted time axes are confusing, which has also been pointed out by the second reviewer. We have modified the figure to show that the time axes are aligned (Figure 4 in this document).

Line 147: Why not extrapolate the values from the centres to the cell boundaries where needed, to minimise the regions of missing data? Would this change anything, or is it implicit in your next point about filling the missing regions?

The algorithm covers this in two steps: first regridding between the grids and secondly filling missing values appropriately. The reviewer is right with the interpretation, that extrapolation to cell boundaries and beyond is part of the second step. Splitting this into different steps, gives the possibility to also take basin boundaries into account. This avoids that values from one basin are extrapolated into a different basin (e.g. across the narrow Antarctic Peninsula where the western basins are much warmer than the eastern basins). To make the difference between regridding and extrapolation more explicit, the manuscript has been updated:

First, the three dimensional output fields (temperature, salinity) are remapped bilinearly from the spherical ocean grid to the Cartesian ice grid. Bilinear regridding is chosen to allow for a smooth distribution of the coarse ocean cell quantities on the finer ice grid. Through regridding, regions with missing values on the ice grid increase (e.g., compare the ocean coverage around the Antarctic Peninsula in Fig. 1b and 5a for example). This is a consequence of bilinear regridding, which interpolates values between the cell centers of non-missing source grid cells and not the cell boundaries. Only regions with valid ocean data are filled on the ice grid, which is up to the cell center of the southernmost ocean cell. Areas with missing data need to be filled accordingly (compare grey areas in Figure 5a for example), which is done in the next step. Another option - linear extrapolation into areas with no ocean data coverage by the bilinear regridding scheme - is not applied here as it can lead to unrealistic results.

Line 148: It is unclear whether the missing values are filled as a single block of adjacent missing values, or cell by cell. This would lead to slightly different behaviour in the averaging routine.

We apologise for the ambiguity. All grid cells with missing values in the same basin and vertical level are filled synchronously with the same averaged value of all adjoining valid cells. The manuscript has been changed to describe this more precisely:

Secondly, missing values are filled with appropriate data, namely the average over all existing values that are adjacent to grid cells with missing values. This procedure is conducted for each basin and vertical layer, using the same mean value of adjoining valid cells for all missing grid cells in that region. Now, the continental shelf area between the ice shelf front and the continental shelf break (see Figure 5a), which is used by PICO to calculate the basin mean values of oceanic boundary conditions, is entirely filled with appropriate values.

Line 154: What happens to the vertical interpolation if the ocean bathymetry is shallower than the continental shelf depth as seen by PISM?

In this case, the vertical interpolation is omitted and the algorithm chooses the lowermost ocean layer available instead. This has been added to the current version of the manuscript:

Lastly, the three dimensional variables are reduced to two dimensional PICO input fields which represent the ocean conditions at the depth of the continental shelf. This is done by vertical linear interpolation: for every horizontal grid point, the data is interpolated to PISM's mean continental shelf depth of the corresponding basin. In case the ocean bathymetry is shallower than the continental shelf depth as seen by PISM, the lowermost ocean layer is chosen. An example of the processed input data for PICO is shown in Fig. 5b.

Line 163: If I understand correctly, every coastal ocean cell in a given basin receives the same mass of freshwater from PISM (and similarly for energy). Why is this not scaled by area, so that larger ocean cells receive more freshwater? This would be equivalent to distributing the total mass flux evenly over the coastline in the given region, regardless of the details of the ocean grid.

The reviewer is correct that in the original version of the coupling framework as described in the preprint each ocean cell in the same PISM basin receives the same mass of freshwater and energy. This appeared to be a reasonable choice, as usually most of the ocean cells mapped to a given PICO basin have the same latitudes and thus the same area. However, also different ocean cell areas can occur in the same basins, introducing subtle inconsistencies in the distribution of the mapping algorithm. To avoid this, we have adapted the mapping algorithm to calculate the fraction of attribution for each ocean cell weighted by the cell area. Even though the difference is minor in the presented setup, the mapping is more generic now and robust also for different configurations. We thank the reviewer for pointing out this issue.

Figure 7: It's confusing to compare the different units in (a) and (b). Over what time interval are the fluxes in (a) integrated?

The interval over which the fluxes in Figure 7a have been integrated is a 10 year coupling time step. We apologise that this was not clear and thank the reviewer for addressing this. To enable a better comparison of fluxes between (a) and (b), the labels have been standardised and show the fluxes now both in units  $kg/m^2/s$ . The improved figure including the modified description is shown in this document as Figure 6 (Figure 7 in the preprint).

Line 171: What happens if the energy flux causes supercooling of the ocean waters? Is this supercooling automatically removed by the sea ice model?

This is correct. The energy flux from ice sheet to ocean component is typically negative, which results in a negative enthalpy flux applied to the ocean component. When this causes a drop of temperature below the local freezing point, frazil ice is formed by the sea ice model.

Line 175: Doesn't ignoring these heat fluxes mean the coupled model does not conserve energy? I understand the argument later that the coupler itself conserves energy, during the regridding process, but it appears false to claim that the entire system is conservative (as is implied in line 317).

Indeed the statement that 'conservation of mass and energy is obtained in the coupled system' (l.317) is misleading when referring to the coupling interface only. This work does not and cannot aim at improving the performance or the conservation in the coupled model components. It aims (only) at not introducing additional instabilities or conservation errors in the coupling process. We thank the reviewer for spotting this. The manuscript has been modified accordingly:

We can assure that conservation of mass and energy is obtained in the coupled system coupler between the ocean and land ice components while having a computationally efficient and flexible coupling setup.

Line 188: Explain why the coupling time step is an important parameter. What do you expect would happen if it were too long or too short?

We agree that it should be made clearer why the coupling time step is important and thank the reviewer for pointing this out. A too short coupling time step certainly results in a waste of compute time and disk space for restart and coupling overhead. Too long coupling time steps on the other hand could possibly yield instabilities and might not capture the physical interactions between the components adequately. So far, the manuscript has been modified as follows:

For the modelling of ice-ocean interactions, the coupling time step is an important parameter that requires careful adjustment, while keeping the different time scales of ice and ocean processes in mind. In order to assess Too short time steps certainly yield a waste of compute time and disk space for restart and coupling overhead. Too long time steps could possibly yield instabilities and lead to a less accurate representation of ice-ocean interaction processes. Here, only the influence of the coupling frequency on the overall performance, two-runtime performance is assessed, leaving the examination of physical implications to Section 5.3. Two experiments with time steps of 1 and 10 years are compared. The experiments have, with a total number of 200 and 20 coupling iterations, respectively, and the individual coupled model. The individual coupled component simulations start from quasi-equilibrium conditions.

As described above (major comment on model validation), we have included a new Section (5.3 Coupled runs for present-day conditions). A comparison between two runs was made over a time period of 4000 years. While both runs are using a coupling time step of 10 years, one setup provides the mean ocean forcing over the coupling time step to the ice model, while the other uses a timeseries forcing of annual averaged ocean temperature and salinity and thus reflects the ocean forcing variability of a yearly coupling time step. No significant difference was observed between both runs.

Line 200: Clarify that MOM5 runtimes are slightly longer due to a greater fraction of time spent initialising, as with PISM (I assume this is the case).

Yes, the subtle increase in MOM5 execution times is due to the model initialisation, which is done 10 times as often in the yearly coupling compared to the decennial coupling setup. This has been clarified in the manuscript, which also includes an update of the benchmark numbers:

In the experiment using a yearly coupling time step, total the elapsed time for all MOM5 execution times executions increases slightly ( $13\ 28015\ 446\ s$ ) compared to 10 yearly coupling ( $12\ 330267\ s$ ). The ocean model increase is due to component initialisation overhead which occurs 10 times as often as in the decennial coupling configuration. The ocean component postprocessing (9%) and intermodel intercomponent processing routines (64%) are taking a bigger share of the total runtime as they are invoked 10 times as often as in the decennial coupling configuration, the number of executions has similarly increased by the factor 10.

Line 222: Does this statement mean that MOM5 does not conserve mass? That is concerning, and should be explained further and referenced.

The purpose of this work is to show that there are no mass conservation errors introduced in the coupling process. As we show here, indeed, errors in the coupled model are minimal. However, there seems to be a spurious mass drift in the specific coarse-grid MOM5/SIS *standalone* setup used in our experiments. While we agree with the reviewer that this needs to be investigated further, it would be beyond the scope of this study.

Line 250: As the Galton-Fenzi reference only refers to an EGU presentation, a better reference for online coupling would be Jordan et al. 2018 (doi:10.1002/2017JC013251) which is already published in JGR.

We thank the reviewer for this suggestion. This is indeed a better reference than the EGU presentation. The reference has been changed in the manuscript.

Line 257: I'm not sure it's essential for online coupling to have the same timestep for ice and ocean, as ocean models often subcycle different timesteps for different processes (eg barotropic vs baroclinic modes).

Indeed, in a coupled model there are generally several processes running on different time steps inside the different model components. One example, as pointed out by the reviewer, are the barotropic/baroclinic time steps inside MOM5. It would be possible to introduce an additional outer time stepping cycle which wraps the ocean and sea-ice time steps to synchronise to with an online coupled land ice component. But that would require subsequent modifications in the code structure of coupler, ocean, and land ice model, which we can avoid with the offline coupling approach. Furthermore, PISM's adaptive time stepping scheme with all its advantages could hardly be used in that approach.

Line 273: The ocean mixed layer will rarely extend as deep as the ice shelf front. Ice shelf meltwater entering the ocean at depth would therefore destabilise the water column and deepen the mixed layer, whereas applying this flux at the surface would have the opposite effect. I appreciate that applying meltwater at depth is not trivial in MOM5, but more attention should be given to the possible negative impacts of this design choice on the ocean simulation.

We thank the reviewer for raising this point and agree that the consequences of meltwater input depth needs more consideration. According to Pauling et al. (2016), the spread of freshwater input depth ranges from (close to) surface up to 500 m in depth (derived from RTopo-1 dataset, Timmermann et al. 2010). In their simulations a seasonal dependence of whether the input depth is within in the mixed layer or below (especially in summer and autumn) is observed (Pauling et al., 2016, Fig.4). To consider the shortcomings of surface meltwater input in our study in more detail, we have modified the manuscript as follows:

As described in Section 4.2, the mass and energy fluxes computed from PISM output are given as input to the ocean surface rather than being distributed throughout the water column - a limitation of MOM5. MOM5's simplified treatment of all land-derived mass fluxes, including those from ice sheets. This simplification may affect vertical heat distribution and local sea ice formation (Bronselaer et al., 2018) as near-surface input generally makes the vertical column more stratified, whereas input below the mixed layer destabilises the water column, thus enhances vertical mixing and extends the mixed-layer depth (Pauling et al., 2016). A more realistic input depth into the ocean would be the lower edge of the ice shelf front (see start of upper green arrow in Fig. 3; Garabato et al., 2017) which could be determined as the average ice-shelf depth of the last PICO box. However, considering the turbulence in the ocean mixed layer, the simplification of surface input seems reasonable for most cases.

**Comments by Reviewer 2 (Rupert Gladstone)**

The paper describes an implemented approach to interactive coupling between an ice sheet model and an ocean model through a reduced complexity ocean cavity model. The project provides a compromise between complexity and efficiency, allowing large scale coupled simulations in which at least some aspects of cavity circulation are represented. This is a useful contribution to global modelling efforts. For the most part, the work is very clearly described and presented. The figures are well prepared and appropriate. I would recommend publication of this paper with some fairly minor modifications.

We thank Rupert Gladstone for taking the time to review our manuscript. The comments are very helpful and we will answer them point by point below.

**General Comments**

I'm not entirely sure what other models might be competing with the current study in terms of global coupled models of intermediate complexity, but I think maybe the UKESM falls into this category? Are there publications about UKESM and how does it compare to your approach? Perhaps it is significantly more computationally expensive than your setup? I think it uses BISICLES and NEMO for ice and ocean models.

Thanks for raising the point of comparable existing work. This had not been sufficiently addressed in the preprint-version of the manuscript.

There is ongoing effort on coupling the ice sheet model BISICLES to the ocean model NEMO as part of the UKESM development, referred to as UKESM-IS. Unfortunately, we could not find any publication describing more details about the coupled model in general, and especially how the coupling is implemented. A recent study by Gierz et al. (in review, 2020) describes the coupling of AWI-ESM with PISM via SCOPE which is using a comparable approach to our offline coupling framework, although focusing on ice-atmosphere coupling on the Greenland Ice Sheet. The revised introduction includes now a general overview of existing ice-ocean coupling approaches and comparable work (see also reply to first major comment of reviewer 1 above).

The first reviewer recommended including model validation in this paper. Robust model validation is a very large challenge and is not typically included in model description papers. I would prefer to see separate studies present some level of validation rather than try to include it here. Note that this is a highly tunable model, and so a good match to observations over a short time period should be straight forward to achieve. The real challenge will be in quantifying the uncertainty in the model as conditions evolve significantly over long periods of time, and this challenge is well beyond the scope of the current paper. The paper does present some level of model verification, especially regarding conservation, and this is useful.

We agree with the reviewer that fully validating the model would require assessing uncertainties in simulations over long time periods. So far, we have included a new Section in the manuscript (5.3 Coupled runs for present-day conditions, see also response to first major comment by reviewer 1), which shows the spin-up procedure of the coupled setup and that the coupled setup is able to simulate a stable ice sheet and ocean under present-day climate. More detailed analysis and model validation will then follow in a subsequent publication.

I am aware that some aspects of my review read like an advertisment for the Earth System Modelling Framework (ESMF)! I shoud clarify I am fully independent of ESMF; I'm just an end user. In my experience ESMF is a very robust, user-friendly, well-documented code. So I guess I'm just a fan! It is a valid point though. Existing coupling frameworks (and I talk about ESMF rather than OASIS or others simply because I have the most experience with it) do provide more features and flexibility than the authors convey knowledge of in some places (see also my specific points below).

Thanks for pointing us to these frameworks. See also the replies to the comments below.

It is very interesting to learn that the PISM intialisation time becomes a significant factor as coupling timestep is shortened. It makes the question "what coupling timestep do you need?" rather important. I couldn't find a comparison of important result metrics between the 1 year and 10 year coupling timesteps. Information about computational cost is given, but is the actual behaviour different? For example, do they give the same total melt rate over the 200 years? Does the ice shelf thickness evolution look the same in both simulations? And grounding line and ice front evolution? You can't really address whether your framework meets requirement 2 until you've established whether or not a 10 year coupling timestep is sufficient.

This is a very valid point and in line with the first reviewer's comment on model validation and coupling time step (line 188). Thanks for addressing this. As mentioned in the response to the comments of the first reviewer, we have compared two runs over a time period of 4000 years. While both runs are using a coupling time step of 10 years, one setup provides the mean ocean forcing over the coupling time step to the ice model, while the other uses a timeseries forcing of annual averaged ocean temperature and salinity and thus reflects the ocean forcing variability of a yearly coupling time step. As no significant difference was observed between both runs, we conclude that a coupling time step of 10 years is sufficient to produce reasonable results and that subsequently requirement 2 (computationally efficient coupling) is fulfilled. The discussion has been extended slightly as follows:

Compared to the required run time of MOM5, the framework routines are very efficient when choosing a reasonable coupling time step of 10 years. More frequent coupling causes a larger overhead, as reading and writing the complete model state of PISM to and from files is relatively expensive for very short simulation times. However, an increased ocean to ice forcing of 1 year does not affect the equilibrium state of the coupled system as shown in Section 5.3.

I didn't find information about parallelism of the coupling. I infer from this lack of information that the coupling (bash script and file manipulation) all occurs on one processor. If this is the case, can you comment on how many processors you intend to use for production runs, and whether this might become a bottle neck for larger parallel simulations? For example, the ocean postprocessing and intermodel processing took 15

Currently the data processing for the coupling is implemented sequentially. Compared to the compute times of component models it is so small that it appears not worth the effort to parallelise it. For production runs we are currently using 32 cores. For higher grid resolutions, the interpolation processing similar to the integration time of the components will require longer run times. We do not expect the interpolation to become a bottleneck in this case (in particular since we rely on bilinear regridding and not conservative regridding which is much more expensive). If it becomes a bottleneck, we agree that this would be a good way to improve performance.

The Earth System Modelling Framework (ESMF) community adopts some terminology that I find quite beneficial in that it offers clarity in certain areas that can otherwise become slightly confusing. Individual models in a coupled system are referred to as components. This avoids confusion between the coupled model and the individual ice and ocean models. I like this clarity and I think it would be nice the the authors adopt it, but I do not wish to make this a requirement, just a suggestion.

We agree that addressing the coupled setup as 'model' and the individual ice and ocean submodels as 'components' introduces more clarity. We thank the reviewer for this suggestion and updated the manuscript accordingly.

**Specific Comments**

Line 107: I don't quite understand the use of two horizontal dimensions. If I understood right, PICO just represents the overturning circulation. How do two horizontal dimensions come into play here? Or perhaps I should go and read Ronja's 2018 paper on PICO (lazy reviewer!)!

The implementation of PICO includes a routine to process the 2-dimensional input fields. It averages the input over the continental shelf in front of the ice shelf cavity in each basin. This makes it easier to have changes in the ice front position feed back onto the ocean forcing during the transient simulation. The input for the box model solutions is then a single value in each ice shelf. To make this a bit clearer, we changed "required" to "uses" in the manuscript:

PICO requires uses two dimensional (horizontal) input fields, namely temperature and salinity of water masses that access the ice-shelf cavities, to calculate melting and refreezing at the ice-ocean interface, as illustrated in Fig. 3.

Lines 113-114: This is also known as "sequential coupling".

Thanks for this remark, it has been added to the manuscript:

This is one motivation among others to use an offline sequential coupling approach to exchange the fields between the two components.

Line 115: In this context, does an "integration step" mean running the model for a coupling time step? Can you clarify this in the text?

Exactly. The term "integration step" has been used as a synonym for "coupling time step" in this context. To avoid confusion, this has been changed and clarified in the manuscript:

During offline coupling, the model output after each model integration step is processed and provided as input or boundary condition to the other model, respectively. In the offline coupling procedure, one component is first run for the period of a coupling time step. Then the output is processed and provided as input or boundary condition to the other component, respectively. Using the modified input, the models are restarted from their previous computed state.

Line 118: "the last of these averaged fields" is a slightly confusing expression. Presumably PISM receives the average of the fields over the full (10 years in this example) coupling timestep? But this isn't clear to me from the chosen wording.

In the initial implementation of the framework as described in the preprint, PISM received the last record of the annual mean diagnostic output for ocean temperature and salinity. We have changed the implementation to pass ocean temperature and salinity to PISM that are averaged over the full coupling time step of the last ocean model execution, or optionally the annual time series forcing. We thank the reviewer for raising this point. The changes in the manuscript are the following:

For example, MOM5 runs for 10 years and writes annual mean diagnostics fields of temperature and salinity. PISM receives the last of these averaged fields temporal average of these fields over the coupling time step as boundary conditions for PICO, and is then integrated for the same 10 year period. Melt water and energy fluxes derived from PISM output are aggregated over the coupling time step. The resulting fluxes are then added as external fluxes to the ocean over the course of the next integration period. To avoid shocks in the forcing, they are distributed uniformly over the entire coupling time step.

Figure 4: There appears to be a temporal offset between  $t_{ice}$  and  $t_{ocean}$  in the way that the Figure is presented. But the text suggests that the time period over which the two components are integrated should not be offset (line 113 refers to the "same model time"). Can you clarify this? It is not clear to me what is meant be "Sharing the same time axis", perhaps this is related to my question?

In our implementation, the model components share the same time axis. We admit that this can be understood differently through the way it is presented in Figure 4 of the preprint. The same point was also raised by the first reviewer, and we have modified the figure to make clear that the time axes are aligned (Figure 4 in this document).

Line 134: I don't know what "entanglement" means in this context.

We acknowledge that the chosen wording was not clear and thank the reviewer for pointing this out. By "horizontal grid entanglement" we mean the way that both horizontal grids are intertwining such as there exists overlaps as well as spatial gaps of both model grid domains. The corresponding text has been rephrased in the manuscript:

The ice and ocean models components operate on independent, non-complementary computational grids. The inset of Fig. 3 shows that there are both, spatial gaps and overlaps, between the ocean grid cells and the ice extent represented by PISM. As the ocean grid is much coarser than the ice grid and MOM5 cells are either defined entirely as land or ocean (no mixed cells allowed), inconsistencies in the horizontal grid entanglement exchange of quantities between the two grids are unavoidable, requiring careful consideration of data exchange. The grid remapping mechanisms presented in the following sections are independent of the used grid resolutions data regridding. Line 139: Is "surrounding ice sheets" a typo in this context? I mean, the Zwally basins divide up the Antarctic Ice Sheet... perhaps you mean shelves not sheets here?

This is a typo in the text. We thank the reviewer for spotting this. The text has been corrected accordingly:

They are based on Antarctic drainage basins defined in Zwally et al. (2012) and extended to surrounding *ice shelves* and the Southern Ocean, see Fig. 5b.

Lines 148-149: If you use only adjacent cells to populate missing values, presumably you iterate until all cells have a value? I mean there must, to start with, be plenty of cells that are not adjacent to a cell with a value.

We apologise for the ambiguity. All grid cells with missing values in the same basin and vertical level are filled synchronously with the same averaged value of all adjoining valid cells. The manuscript was modified to describe this more precisely:

Secondly, missing values are filled with appropriate data, namely the average over all existing values that are adjacent to grid cells with missing values. This procedure is conducted for each basin and vertical layer, using the same mean value of adjoining valid cells for all missing grid cells in that basin. Now, the continental shelf area between the ice shelf front and the continental shelf break (see Figure 5a), which is used by PICO to calculate the basin mean values of oceanic boundary conditions, is entirely filled with appropriate values.

Lines 159-160: I'm fairly confident that ESMF's "common" regridding algorithms include masked nearest neighbour remapping options that are very similar to what is described here. I intend to use these for remapping subglacial outflow from an ice model to an ocean model, though I haven't actually implemented this yet, and it looks like the required functionality to do this in a mass conserving way is already in place.

Thanks for pointing out that the ESMF regridding tools support our use case. By using the PICO basins, we have the advantage that the ocean cells and the PICO forcing match nicely in terms of fluxes distributed in either way. We have made the following changes to the manuscript:

Since There are large areas of the PISM domain that are not overlapping with valid MOM5 ocean cells (see white areas in Fig. 1b and inset in Fig. 3), common regridding algorithms would ignore quantities in those areas and consequently violate. To ensure mass and energy conservation. Thus, we introduce a new mechanism for the coupled system which maps every southernmost ocean cell of the MOM5 grid to exactly one PICO basin (see Fig. 6).

Section 5.1. I can't quite make the numbers add up here. The run with a 1-year coupling timestep takes 22700s. The ocean post-processing, interprocessing and PISM percentages given add up to 35

We understand that confusion is caused by not mentioning all percentages in the text and thank the reviewer for bringing this up. For clarification, we added a table of measured runtimes and their corresponding total percentages in the appendix (shown here in Table 1). In the preprint version, the benchmark did not list the runtime required by the coupling post processing (concatenation of coupling time step output files). This has

| F                          |                 |           |                  |           |
|----------------------------|-----------------|-----------|------------------|-----------|
|                            | 1 year coupling |           | 10 year coupling |           |
| routine                    | time $[s]$      | ratio [%] | time $[s]$       | ratio [%] |
| total                      | 21976.49        | 100.00    | 13244.80         | 100.00    |
| preruns                    | 24.17           | 0.11      | 24.41            | 0.18      |
| preprocessing              | 40.97           | 0.19      | 43.03            | 0.32      |
| MOM runs                   | 15446.26        | 70.29     | 12267.26         | 92.62     |
| MOM postprocessing         | 1993.09         | 9.07      | 205.98           | 1.56      |
| PISM runs                  | 2830.57         | 12.88     | 467.26           | 3.53      |
| MOM-to-PISM processing     | 861.89          | 3.92      | 125.43           | 0.95      |
| PISM-to-MOM processing     | 90.43           | 0.41      | 14.01            | 0.11      |
| concatenating output files | 656.44          | 2.99      | 81.91            | 0.62      |

Table 1: Runtimes of coupled PISM-MOM5 setup for 200 years model time using 32 cores. PISM runtimes include PICO and MOM5 runtimes include SIS and FMS components.

been corrected and the benchmark numbers have been updated accordingly.

Line 193: I presume "total runtime" is the elapsed time not the cpu hours? i.e. the total computational time would be this number times 32?

The reviewer is right with his interpretation. With "total runtime" we are referring to the wall-clock time or elapsed time. CPU hours, if defined as the sum of elapsed time on every allocated CPU cores, would be the wall-clock time times 32 in this case. The manuscript has been changed to describe more precisely what we mean with 'total runtime':

The elapsed total runtime (wall-clock time) required for 200 years model time is 22700s and 13700s with a coupling time step of 1 and 10 years, respectively.

Equation 1: Surely mass has dimensions of mass and smb has dimensions of mass/time. So how can you simply add these? I don't understand why there would be rates in this equation. Surely the total mass is the sum of the mass from each component at any given moment in time? The same comment for lines 214-215. How can you subtract a flux from a mass? I can't make sense of it!

Line 222: It is not clear from this description whether the dimensions of dosi should be mass or mass/time.

We admit that the chosen expressions are miss-leading and apologize for the confusion. We renamed  $smb_{osi}$  and  $smb_{li}$  to  $m_{osi}^s$  and  $m_{osi}^s$  respectively as they are not describing the actual mass balance (mass/time) but the cumulative, integrated surface mass balance flux over space and time of the ocean-sea ice component and the land ice component, which is a mass. The same applies to the model mass drift, which was renamed from  $d_{osi}$  to  $m_{osi}^d$  as this is also a mass. The manuscript was changed accordingly:

To check that the total amount of mass and energy stocks is constant in the coupled system over the model integration, we assess virtual quantities. Those are obtained by subtracting the masses applied through surface fluxes from the total mass and energy stocks calculated in the model (see Eq. (1) for mass). If the virtual model mass across the model domains components  $m_v$  is constant with fluctuations in the order of

machine precision, as denoted in Eq. (2), conservation of mass is achieved.

$$m_{\rm v} = (m_{\rm o} + m_{\rm si} - \underline{smb}\underline{m}_{\rm osi}^{\rm s} - \underline{d}\underline{m}_{\rm osi}^{\rm d}) + (m_{\rm li} - \underline{smb}\underline{m}_{\rm li}^{\rm s})$$
(1)

$$\frac{d}{dt}m_{\rm v} \sim 0\,Gt/a\tag{2}$$

The masses of the ocean, sea ice and land ice models components are represented by  $m_o, m_{si}$  and  $m_{li}$  respectively, while  $smb_{osi}$  and  $smb_{li}$  denote the  $m_{osi}^s$  and  $m_{li}^s$  denote the cumulative, spatially integrated surface mass balance flux of the ocean-sea ice model component MOM5/SIS and the land ice model component PISM, respectively. The internal model drift of mass in the coarse-grid MOM5/SIS setup is described by  $d_{osi} m_{osi}^d$  ( $\approx 4 \cdot 10^{15}$  kg in accumulated over 200 years) and needs to be considered in the computation of virtual model mass in Eq. (1). The absolute and All terms in Eq. (1) are quantities of mass with the temporal resolution of the coupling time step. The relative mass conservation errors are calculated according to Eq. (3) and error  $e_{rel}^m$  is calculated as fluctuations of the virtual model mass compared to its temporal mean  $\overline{m_{u}}$ , noted in Eq. (5), respectively.

$$e_{\rm abs}^{\rm m} = m_{\rm v} - \overline{m_{\rm v}} \tag{3}$$

$$e_{\rm rel}^{\rm m} = e_{\rm abs}^{\rm m} / \overline{m_{\rm v}} \tag{4}$$

$$e_{\rm rel}^{\rm m} = \frac{m_{\rm v} - \overline{m_{\rm v}}}{\overline{m_{\rm v}}} \tag{5}$$

Figure 9: To put this apparently small error into context, it would be useful to give some indication of how much mass is transferred between ice and ocean. I think the total mass transfer is probably a more relevant figure here than the total ice or ocean mass. We need to know that the error measures are small not only compared to the total mass of the coupled system, but also small compared to the amount of mass being transferred between ice and ocean over the integration period.

We thank the reviewer for this very good suggestion. A subgraph has been added to Figure 9 (Fig. 7 in this document) which compares the fluctuations of the virtual coupled mass to the mass transferred between the components. The text and the figure's caption has been changed accordingly:

The relative mass conservation error  $e_{rel}^m$  (see Eq. (5)) is shown in Fig. 7a for 200 model years with a yearly coupling time step. Regarding the order of magnitude of land ice mass  $\mathcal{O}(m_{li}) = 10^{19}$  kg which is given in single precision ( $\approx$  7 decimal digits) output format and the order of magnitude of ocean and sea ice mass  $\mathcal{O}(m_o + m_{si}) = 10^{21}$  kg, given in double precision ( $\approx$  16 decimal digits) format, the shown fluctuations in the order of  $10^{-9}$  are reasonable. As the relative mass error does not show a trend, no systematic error is introduced through the coupling procedure. In Fig. 7b the fluctuations of virtual model mass is also compared to the mass flux between the land ice and ocean component  $(m_x)$ , which is in the order of  $\mathcal{O}(10^{-3})$ .

Line 239: I don't think you can make this unqualified statement that the framework fulfills all three requirements. 1 and 3, yes, sure, but requirement 2 is really only partially fulfilled, as you continue on to discuss in the following paragraph.

We thank the reviewer for the comment, and agree that this statement was not sufficiently proven in the preprint. As we can't observe significant differences between using the 10 yearly mean or yearly time series forcing (see comments above and the new Section 5.3 Coupled runs for present-day conditions), we conclude that requirement 2 is fulfilled. The discussion has been extended slightly:

The framework presented here to couple the ice model component PISM to the ocean model component MOM5 via PICO fulfills all three goals stated in the Introduction, which are (1) mass and energy conservation across both model component domains, (2) an efficient as well as (3) generic and flexible coupling framework design: As described in Section 5.2, mass conservation across both model component domains can be assured. Furthermore, the remapping scheme for energy fluxes is conservative as well. Compared to the required run time of MOM5, the framework routines are very efficient when choosing a reasonable coupling time step of 10 years. More frequent coupling causes a larger overhead, as reading and writing the complete model state of PISM to and from files is relatively expensive for very short simulation times. However, an increased ocean to ice forcing of 1 year does not affect the equilibrium state of the coupled system as shown in Section 5.3. The third criterion is fulfilled by the chosen offline coupling approach, which provides a generic and flexible design by making use of the model-related component-related flexibility concerning grid resolution and degree of parallelisation.

Paragraph 248-267. This paragraph completely omits to discuss alternative forms of online coupling in which a single executable links to component runtime libraries and fulfills a role equivalent to your bash script as a master (or parent) program. The implication in this paragraph is that one must choose between an offline coupling in which components are called independently and an online coupling in which one component must be the master and the other the slave. But this is not the choice that a coupled model developer faces, there are many more options available. For example, the Earth System Modelling Framework (ESMF) offers full flexibility in this sense. The developer can choose whether to create a new parent routine or to establish one component as master. You might want to cite Gladstone et al GMDD FISOC paper (published in June 2020 as a discussions paper and now accepted pending minor revisions for GMD), which is essentially an online equivalent of your coupling structure, with a new (Fortran in this case) master program that calls the child components (which have been made ESMF compatible and compiled as libraries). This approach also allows flexibility in terms of switching between components (indeed we currently have a choice of two ocean models coupled through FISOC, and two further ice sheet models are in the process of being incorporated). Of course there is still some overhead in terms of ensuring each component is compatible with the coupling framework (ESMF in the case of FISOC), so it is a longer development path than your bash script plus file manipulations, but perhaps not as onerous or restrictive as implied in the current version of the text.

We thank the reviewer for the remark. We appreciate the possibilities of ESMF and FISOC architectures and are happy to learn more about them in the future. The variant proposed by the reviewer (new main program which as acts as coupling master and links to component runtime libraries) has mostly the same disadvantages as using one of the components as master and the other as slave, linked together into one online coupling executable. In both cases it is necessary to modify at least one of the component model codes in a suitable way so that it can be linked into the executable program. There might be model codes for which such modifications were already done, or anticipated, by others before, and thus are easier to link into a common executable. In fact, MOM is prepared to be linked into coupler frameworks like FMS or Access, but PISM is not. In our work, we preferred to avoid writing a new main program and maintaining interfaces altogether, by using the described approach via a master shell script, with all the described advantages. To include the reviewer's reasonable argument about different possibilities for online coupling architecture (master-slave structure), we have made the following changes to the manuscript:

The chosen offline coupling framework executes the two different components alternately and independently and takes care about redistributing the input and output files across the components as explained in Section 3. However, it is also conceivable to adopt an online coupling approach (also called synchronous coupling), where the ice and ocean component code are consolidated into one code structure (Jordan et al., 2018). Consequently, the. The exchange of variables of between both components can subsequently take place through access to the same shared memory instead of writing the required variables to disk and reading from there again, as it is done in offline coupling. The downside is that a potential integration of PISM into the existing code structure of This approach is for instance used by Jordan et al. (2018). A comprehensive framework for online coupling of ocean and ice components is described in Gladstone et al. (2021). This coupling approach is especially powerful for high resolution, cavity resolving ice-ocean coupling, where frequent updates of the ice-shelf cavity geometries and corresponding melt rates are important. However, a prerequisite for online coupling is the adaptation of the ocean component MOM5 and its driver would require heavy standalone models for interactive execution of subroutines through a defined (external) interface. In the given case of coupling PISM and MOM5, this means that at least one of the two programs' code structure needs major modifications and modularisation of the PISM main routine which is responsible for model initialisation, the time stepping routines to equip the individual component parts like initialisation, time stepping routine, disk I/O, stock checking, etc. Similarly the ocean component main routine would have to be extended to integrate all relevant PISM parts at the right place including MPI parallel mechanisms for data exchange between the components with suitable interfaces. This is independent of the chosen online coupling design (incorporating one code structure into the other or creating a new master program which governs both components). Synchronisation of the PISM adaptive time step and the fixed ocean component time step would be a further issue, also keeping in mind that the comparably small ocean time step of a few hours is not applicable for the ice component: PISM can have a time step of around 0.5 years close to equilibrium with 16 km resolution due to the longer characteristic timescales of ice dynamics.

Line 256. ESMF (for example) handles all parallel regridding in an efficient manner. So long as component mesh and field information can be made available in ESMF runtime data structures (which I acknowledge requires some coding and may not be trivial), the regridding between different partitioning is all handled automatically.

Thanks for the remark. The referenced sentence has been removed from the manuscript (see comment above).

Line 260. The C/Fortran issue is a fairly small technical issue. There are plenty of codes around that use both at runtime.

We agree, and mention it just for completeness.

Lines 269 - 274. The physical implications of this issue could be fairly interesting. Could the input ever come in beneath the turbulent mixed layer? Could the input of fresh water at the surface have a stabilising effect that would not occur if it was mixing up from lower down? Do you have any plans to investigate this further? This could all raise interesting questions that have been brushed over very lightly here.

This has also been noted by the first reviewer and we appreciate both comments on this issue. According to Pauling et al. (2016), the spread of freshwater input depth ranges from (close to) surface up to 500 m in depth (derived from RTopo-1 dataset, Timmermann et al. 2010). In their simulations a seasonal dependence of whether the input depth is within in the mixed layer or below (especially in summer and autumn) is observed (Pauling et al., 2016, Fig.4). To consider the shortcomings of surface meltwater input in our study in more detail, we have modified the manuscript as follows:

As described in Section 4.2, the mass and energy fluxes computed from PISM output are given as input to the ocean surface rather than being distributed throughout the water column - a limitation of MOM5. MOM5's simplified treatment of all land-derived mass fluxes, including those from ice sheets. This simplification may affect vertical heat distribution and local sea ice formation (Bronselaer et al., 2018) as near-surface input generally makes the vertical column more stratified, whereas input below the mixed layer destabilises the water column, thus enhances vertical mixing and extends the mixed-layer depth (Pauling et al., 2016). A more realistic input depth into the ocean would be the lower edge of the ice shelf front (see start of upper green arrow in Fig. 3; Garabato et al., 2017) which could be determined as the average ice-shelf depth of the last PICO box. However, considering the turbulence in the ocean mixed layer, the simplification of surface input seems reasonable for most eases.

Lines 285 - 288. This is actually a very important issue, especially because you intend this framework to be applied to long timescale simulations. You mention that work is in progress, but can you say a few lines about how this will be implemented? Can you also ive an example or two of the how you intend to use the model in its current form so that the reader can start to envisage how much of an issue (or not) the lack of evolving active ocean domain is?

Thanks for this comment. We have adapted and extended the paragraph to address the request for more details on the described mechanism (see below). For the multi-millennial coupled runs that prove the stability of a coupled configuration for present-day conditions (included in the revised manuscript), the described mechanism is not required due to a lack of significant mass changes between ocean and ice. Future analysis of long term interactions will include a more detailed consideration of this mechanism.

**Changes in the manuscript:**

During long simulations where glacial and interglacial periods are alternating, large amounts of water are transferred between oceans and ice sheets. Through significant changes in the sca level, whole ocean cells can be subject to wetting or drying. The land-ocean mask needs to be adapted accordingly during the simulation including a meaningful way to handle mass and energy stocks. The current framework is not capable of managing such changes yet, but development is currently in progress. The waxing and waning of ice sheets on glacial-interglacial time scales causes transfer of large amounts of water between the oceans and land ice sheets. Significant changes in sea level (120-135 meters below present during the last glacial maximum (Clark and Mix, 2002)) have large impacts on coast line positions. The response of the solid Earth component to changes of ice-sheet mass has a similar effect. During long simulations the land-ocean mask needs to be adapted accordingly. As MOM5 cannot handle mixed ocean-land cells, which would allow for a smooth adaption of a changing coast line, major changes in the land-ocean mask need to be performed during a transient simulation. This requires careful considerations like the initialisation of newly flooded cells and implications concerning mass and energy conservation as well as model stability. The development of a sea-level based dynamic ocean domain adaptation which applies the described changes to new ocean restart conditions is currently under way and will be incorporated in the described coupled setup in the future.

Lines 293-295. These lines seem to imply that tuning PICO will somehow make up for the lack of a full representation of the complex 3D ocean circulation over the continental shelf and under ice shelves. This will not, in general, be the case. It is clear that this model is a compromise approach, a model of intermediate complexity with the benefit of efficiency. This has value; you don't need to try too hard to defend or justify limitations of PICO. I would prefer to see the limitations presented directly without implying that they can be overcome (through tuning for example).

We thank the reviewer for this thoughtful comment and have modified the manuscript as follows:

In the coupling framework, ocean input for PICO is averaged over the entire basin, not taking into account horizontal differences such as cavity in- and outflow regions and possible modifications of water masses on the continental shelf. Furthermore, complex processes determine what water masses make their way from the open ocean onto the continental shelf and to the grounding lines (Nakayama et al., 2018; Wåhlin et al., 2020). However, in our coarse grid setup of MOM5, bathymetry and circulation on the continental shelf are only partly represented (see also Fig. 1b). PICO currently does not represent circulation patterns besides the vertical overturning circulation, and they hence need to be considered in the tuning process of the PICO parameters. The presented coupling framework is characterized by a reduced complexity approach. This is reflected for instance in the basin wide averaging of PICO input which does not account for horizontal differences such as cavity in- and outflow regions or modification of water masses on the continental shelf. Similarly, the complex processes determining whether upwelling Antarctic Circumpolar Deep Water reaches the continental shelf and the grounding lines (Nakayama et al., 2018), can only be partly represented due to the coarse bathymetric features of the MOM5 grid (see also Fig. 1b). However the intermediate complexity of the coupled system enables ocean simulations on a global domain, opening possibilities to study interactions, feedbacks and possible tipping behaviour on millennial time scales.

311. "appropriate" is not well defined in this context, and perhaps not yet fully justified. How about just "intermediate" instead?

We agree that this is a better phrasing and have changed the manuscript accordingly:

Overall, despite the limitations discussed above, the coarse grid setup of MOM5 in combination with the representation of the ice pump mechanism in PICO, makes large-scale and long-term ice-ocean coupling possible at an appropriate intermediate level of complexity.